# Development and validation of a scoring system to predict mortality in patients hospitalized with COVID-19: A retrospective cohort study in two large hospitals in Ecuador

Iván Dueñas-Espín[1]*, María Echeverría-Mora[1], Camila Montenegro-Fárez[1], Manuel Baldeón[2], Luis Chantong Villacres[3], Hugo Espejo Cárdenas[4], Marco Fornasini[2], Miguel Ochoa Andrade[5], Carlos Solís[3]

1 Instituto de Salud Pública, Facultad de Medicina, Pontificia Universidad Católica del Ecuador (PUCE), Quito, Ecuador, 2 Escuela de Medicina, Facultad de Ciencias Médicas, de la Salud y de la Vida, Universidad Internacional del Ecuador, Quito, Ecuador, 3 Hospital General Norte de Guayaquil, IESS Ceibos, Instituto Ecuatoriano de Seguridad Social (IESS), Guayaquil, Ecuador, 4 Independent Researcher, Ecuador, 5 Hospital General del Sur de Quito, Instituto Ecuatoriano de Seguridad Social (IESS), Quito, Ecuador

☯ These authors contributed equally to this work.
* igduenase@puce.edu.ec

## Abstract

### Objective

To develop and validate a scoring system to predict mortality among hospitalized patients with COVID-19.

### Methods

Retrospective cohort study. We analyzed 5,062 analyzed hospitalized patients with COVID-19 treated at two hospitals; one each in Quito and Guayaquil, from February to July 2020. We assessed predictors of mortality using survival analyses and Cox models. We randomly divided the database into two sets: *(i)* the derivation cohort (n = 2497) to identify predictors of mortality, and *(ii)* the validation cohort (n = 2565) to test the discriminative ability of a scoring system. After multivariate analyses, we used the final model's β-coefficients to build the score. Statistical analyses involved the development of a Cox proportional hazards regression model, assessment of goodness of fit, discrimination, and calibration.

### Results

There was a higher mortality risk for these factors: male sex [(hazard ratio (HR) = 1.32, 95% confidence interval (95% CI): 1.03–1.69], *per* each increase in a quartile of ages (HR = 1.44, 95% CI: 1.24–1.67) considering the younger group (17–44 years old) as the reference, presence of hypoxemia (HR = 1.40, 95% CI: 1.01–1.95), hypoglycemia and hospital hyperglycemia (HR = 1.99, 95% CI: 1.01–3.91, and HR = 1.27, 95% CI: 0.99–1.62, respectively) when compared with normoglycemia, an AST–ALT ratio >1 (HR = 1.55, 95% CI: 1.25–1.92), C-reactive protein level (CRP) of >10 mg/dL (HR = 1.49, 95% CI: 1.07–2.08), arterial pH <7.35

**Data Availability Statement:** All relevant data are within the paper and its Supporting Information files.

**Funding:** IDE, MEM and CMF were supported by Pontificia Universidad Católica del Ecuador (Pontifical Catholic University of Ecuador, https://www.puce.edu.ec/) grant number QINV0318-IINV533020000. The funders had no role in study design, data collection and analysis, decision to publish, or preparation of the manuscript.

**Competing interests:** The authors have declared that no competing interests exist.

(HR = 1.39, 95% CI: 1.08–1.80) when compared with normal pH (7.35–7.45), and a white blood cell count >10 × $10^3$ per μL (HR = 1.76, 95% CI: 1.35–2.29). We found a strong discriminative ability in the proposed score in the validation cohort [AUC of 0.876 (95% CI: 0.822–0.930)], moreover, a cutoff score ≥39 points demonstrates superior performance with a sensitivity of 93.10%, a specificity of 70.28%, and a correct classification rate of 72.66%. The LR+ (3.1328) and LR- (0.0981) values further support its efficacy in identifying high-risk patients.

## Conclusion

Male sex, increasing age, hypoxemia, hypoglycemia or hospital hyperglycemia, AST–ALT ratio >1, elevated CRP, altered arterial pH, and leucocytosis were factors significantly associated with higher mortality in hospitalized patients with COVID-19. A statistically significant Cox regression model with strong discriminatory power and good calibration was developed to predict mortality in hospitalized patients with COVID-19, highlighting its potential clinical utility.

## Introduction

World Health Organization (WHO) has reported a global total of 767,984,989 confirmed cases of COVID-19 caused by Severe Acute Respiratory Syndrome Coronavirus 2 (SARS-CoV-2), with 6,943,390 deaths. Additionally, as of 13 June 2023, a total of 13,397,292,784 vaccine doses have been administered worldwide [1]. In 2021, hospitalizations from COVID-19 have increased worldwide. COVID-19 hospitalizations reached a record high of more than 142,000 on January 14, 2021, and last topped 100,000 on September 11 [2], with a greater rate of hospitalization and death among patients with comorbidities [2], such as obesity [3], and other specific clinical conditions [4]. Moreover, diabetes has been more prevalent among fatal cases than non-fatal cases [5].

Although the mortality rate of patients with COVID-19 in the general population has been reduced by the introduction of vaccination, causing important reductions in infections, hospital admissions, and deaths [6], there is still a large burden of death from the infection [7].

Globally, the vaccination process has been carried out unequally, irregularly, and very slowly. In 2022, 90 countries have not yet reached the goal set by the World Health Organization for 2021 of vaccinating 40% of their population, and 36 countries have inoculated less than 10% of their population [8, 9]. Currently, and despite that 69.7% of the world population has received at least one dose of a COVID-19 vaccine, 13.31 billion doses have been administered globally, and 954,258 are now administered each day, only 27.7% of people in low-income countries have received at least one dose [10]. The slow process of vaccinating could have contributed to virus mutations, resulting in new variants. Specifically, despite Delta variant caused more severe, the extremely contagious Omicron variant has resulted in a new collapse of public and private healthcare facilities increasing the risk for morbidity and mortality of the population.

Importantly, despite the evidence that vaccination may make breakthrough cases less severe and people who are fully vaccinated may be less likely to spread the disease to others, the rate of hospitalization in the first months of 2022 was even higher than that reported during the year 2020 or 2021 [10].

Therefore, it is necessary to estimate factors that can predict the risk of death among hospitalized patients [11]. In addition, it is important to consider that the rise of new variants is causing impacts in terms of public health that have not been seen since 2020 [12]. These variants have increased the number of hospitalized patients throughout the region, and specifically in Ecuadorian territory.

In Ecuador, given the sanitary collapse that occurred between March and May 2020, there is evidence of underreporting COVID-19 deaths in infographics from the Ministry of Health [13]. Our country had one of the highest mortality rates from COVID-19 in Latin America in 2020 and 2021 [14, 15]; currently, 34,533 people have died due to COVID-19 in the country [16]. More than 46,656 excessive deaths have been registered compared to the years prior to the pandemic, giving an excess mortality rate of 266 per 100,000 population [16].

At present, despite the Ecuadorian prevalence of compliance with the complete vaccination protocol is being close to 79.06% (one of the highest in the region) [10, 13, 17], there are difficulties reaching populations where vaccination coverage is complex (*i.e.*, rural areas population), a notable increase in cases has been reported during January 2022 [7]. Among the different possible explanations, the following aspects are the ones that could mainly explain it: *(i)* vaccines are not 100% effective, *(ii)* unvaccinated people could be fueling transmission, *(iii)* spreading the disease by vaccinated cases, *(iv)* people that are moving around more to restart economic activities or during vacation periods and festivities, *(v)* the relaxation of protection measures like distancing measures or the increase in capacity in establishments; and *(vi)* the current global dominance of the SARS-CoV-2 Omicron (BA.1 or B.1.1.529) and several subvariants that are appearing more frequently, with a higher transmissibility, even in vaccinated individuals [18].

With this background, decisions to contain the pandemic should be taken based on proper tools for estimating the risk bands to allocate resources at both clinical and public health levels [19]. This study aimed to identify the prognostic factors associated with in-hospital mortality of COVID-19 from the information of adult patients admitted to two Ecuadorian Social Security Institute Social, the *Hospital General Norte* of Guayaquil and the *Hospital General del Sur* of Quito, between February and July 2020, and to validate and to assess its discriminative ability in the form of a score.

## Materials and methods

### Design

Retrospective cohort study. This is a survival analysis of a retrospectively reconstructed cohort of patients hospitalized for COVID-19 in two large hospitals in Ecuador and developing and validation of a score for predicting in-hospital mortality. The supplementary material containing the TRIPOD [20] statement checklist has been included in the **S1 Table**.

### Population and database

This is an individual-based study whose population is patients over 18 years of age diagnosed with COVID-19 and treated at two large hospitals of the social security system: one each in Guayaquil and Quito, Ecuador, from February to July 2020.

Inclusion Criteria: *(i)* age ≥18 years, *(ii)* confirmed diagnosis of COVID-19 through Reverse Transcriptase-Polymerase Chain Reaction (RT-PCR) testing or findings of computed tomography criteria consistent with COVID-19 [21], *(iii)* hospitalized in one of the aforementioned health facilities, *(iv)* hospitalized from February to July 2020. Exclusion criteria: *(i)* pregnant or in the immediate, early, or late puerperium, *(ii)* primary or congenital immunodeficiency, and *(iii)* history of neoplasms.

### Main outcome

In-hospital death. For the survival analyses, we employed the time from hospitalization to in-hospital death measured in days (survival time). We calculated incidence rates of mortality (mortality rates).

### Explanatory variables

We analyzed several possible explanatory variables as predictors of in-hospital death. We collected information about sex, age, anthropometric measurements, vital signs, and clinical and common-use laboratory biomarkers. We also used the variable hypoxemia. Although there is no specific reference available for the altitude of Quito, efforts have been made to establish a consistent and justified criterion. The operational definition of hypoxemia was based on previous studies, such as the work of Luks and Swenson (2011) [22] and Hupperets et al. (2004) [23], who explored the effects of altitude on respiratory physiology and the hypoxic ventilatory response. In the absence of widely established literature defining Quito's altitude-specific hypoxemia, a consensus was reached based on clinical guidelines and expert opinion [24]. According to this consensus, hypoxemia was defined as an oxygen saturation level below 92% in Quito. Additionally, for sea level (Guayaquil), a threshold below 95% was chosen to define hypoxemia.

### Statistical analyses

The analyses were based on the whole anonymized database of hospitalized patients at both hospitals (Guayaquil and Quito) from February to July 2020. For sample size calculation, based on Cox PH regression with an alpha risk of 0.05, beta risk of 0.2, and a minimum hazard ratio of 1.3, assuming a mortality rate of 14% among patients in the low-risk group [25], it yielded an estimated sample size of 1365 with 191 expected events.

Given that the whole unimputed database had 28.6% of the values missing, we assessed the presence and patterns of missing values and, assuming the missing-at-random hypothesis (*i.e.*, missingness conditional on measured patients' characteristics), we used multiple imputations (10 imputed datasets) with the method of chained equations [26]. We compared the characteristics of chained equations patients in the complete case to the imputed datasets (**S2 Table**). Given that it was not missing values of the in-hospital mortality, except for the analyses of the mortality rates in both hospitals, all analyses were performed using the imputed dataset.

To randomly divide the database into a derivation cohort and a validation cohort, we constructed a sequence of integer values from 1 to 2, as each patient's identifier code was sorted, which divided the entire imputed database in two; therefore, we studied 2,497 patients in the derivation cohort, and 2,565 patients in the validation cohort. Importantly, the number of subjects available was more than twice the required sample, according to sample size calculations.

### Building the model in the derivation cohort

We developed a univariate description of the variables in the first part of the study, followed by a bivariate analysis of in-hospital mortality from the derivation database. Then, we performed multivariate Cox proportional hazards models.

We started by building crude models between one explanatory variable and survival time, which led to the development of a saturated Cox proportional-hazard model including all the variables that achieved a p-value equal to or below 0.25 in the bivariate models. Our saturated model also included variables such as weight, height, body mass index, cardiac rate, diastolic blood pressure, body temperature, ferritin, procalcitonin, platelet count, partial

thromboplastin time (PTT), and D-dimer that did not meet the cutting point but were clinically relevant according to the clinicians' and the researchers' expertise. Subsequently, we built a parsimonious model using a one-to-one variable reduction from the saturated model. It was performed using the researchers' expertise instead of only the p-value as a parameter.

The saturated and parsimonious models were compared, and the final model was chosen according to the p-value of the likelihood ratio test. When the final model was selected, we assessed our model by testing proportional hazards and the overall goodness of fit. The assessment of the Cox proportional hazards model was based on a test of proportional hazards and goodness of fit. We evaluated the goodness of fit through the Cox-Snell residuals graph to visualize a 45-degree angle in the slope to confirm the adequacy of the model and assessed for multicollinearity by using the variance inflation factor.

The model was tested for both goodness of fit and the proportional hazards assumption. Goodness of fit was assessed using the Akaike Information Criterion (AIC) and the Bayesian Information Criterion (BIC) through the "estat ic" command. The proportional hazards assumption was tested with the "estat phtest" command, which involved running chi-square tests for each covariate included in the model.

We stratified the analyses by running the final model in men and women, looking for any possible interaction between sex and other explanatory variables. We also performed a sensitivity analysis by running the final model excluding: *(i)* patients in both extremes of age quartiles, *(ii)* patients with ≥180 mmHg of systolic blood pressure or ≥120 mmHg of diastolic blood pressure; and *(iii)* patients with ≥200 mg/dL of serum glucose.

## Building the score

We used the final model's β-coefficients that were multiplied by a number (20) and rounded to the more approximate integer to build a score. We included the coefficients of the statistically significant variables in the score. Then, we tested the discriminative power of the score, which we called the *COVID-19 in-hospital mortality score*, in the derivation (internal validation) and the validation (external validation) cohorts, by calculating the area under the curve (AUC) in Receiver Operating Characteristic (ROC) curve analyses [27]. To assess if there was a good separation in mortality, we categorized the score as tertiles and assessed if there was differential mortality across risk bands by the log-rank test. For reference, an AUC between 0.7 and 0.8 is considered acceptable, while an AUC between 0.8 and 0.9 is considered to have excellent discriminative power [28].

The categorization scheme for determining risk levels was established using the provided sensitivity, specificity, LR+, and LR- values. The derivation dataset was initially used to identify the cutoff points, and subsequently, the categorization scheme was applied to the validation dataset to evaluate its performance; and then, calculated the survival functions by Kaplan–Meier survival estimates and Log-Rank test.

## Validating the model in the validation cohort

We tested the cutoff value of the score in the validation cohort by calculating the AUC in ROC curve analyses. We used the same reference for assessing the discriminative power. Again, to assess if there was a good separation in mortality, we categorized the score into tertiles and assessed if there was differential mortality across risk bands. We compared the discriminatory power of the score between hospitals and between sex categories, calculating the AUC in the two hospitals and both sex categories (**S3 Table**). These analyses were also performed in the validation cohort only.

## Calibration of the model in the validation cohort

The calibration analysis of the model in the derivation dataset was performed using the Stata command "stcoxcal" [29]. These commands allowed us to assess the calibration of the Cox proportional hazards model by testing the intercepts and slopes against specified values. The "test" command tested the hypothesis that the intercepts are equal to 0 and the slopes are equal to 1, while the "trend" command tested the hypothesis that the slopes are equal to 1 with the intercepts estimated. These analyses were conducted to evaluate the goodness of fit and assess the agreement between the observed and predicted event probabilities, providing valuable insights into the model's calibration performance. Furthermore, we employed the "trend" command in Stata to assess the calibration of the Cox proportional hazards model in the validation dataset. Unlike the "test" command, the "trend" command allows for examining the overall trend of calibration over time.

In the calibration assessment of the Cox model applied to Quito (**S4 Table**), non-significant coefficients were found across all the tests, as evidenced by p-values greater than 0.05. Additionally, chi-square tests further reinforced these findings, with the null hypotheses not being rejected in all cases. These results suggest that the model is well-calibrated for this high-altitude city, as the model predictions do not significantly differ from the observed data. Therefore, the Cox model demonstrates an acceptable predictive accuracy for the city of Quito. Unfortunately, the application of the "stcoxcal" command for the Cox model to Guayaquil was hampered due to the limited number of observations in the original (non-imputed) dataset. This unfortunate circumstance restricts the evaluation of the calibration and predictive model for this city, which underscores the need to collect additional data to provide a more complete understanding of the model's applicability to Guayaquil.

We considered that there were two-tailed statistically significant differences when the p-value was <0.05, and all analyses were performed with Stata 14.2 (*Statistical Software Stata*: *Release 16.1 College Station*, *TX*: *StataCorp LP*).

## Ethics approval

This study was conducted with the authorization of the Ethics Committee of Research on Humans Beings of the Ministry of Public Health, number MSP-CGDES-2020-0183-O, in September 2020. The need for consent was waived by the ethics committee. The information was extracted from patients' clinical records by personnel not related to the research team, who oversaw the systems and technology department of both hospitals. Data were anonymized before being accessed by the researchers.

## Results

A total of 5,062 patients were analyzed. Cumulative mortality during the study was 22.6% (1,139 of 5,062 patients died in six months), which represents an in-hospital mortality rate of 236 deaths per 10,000 person-days (considering a total person-days of follow up of 48,263). Specifically, in the *Hospital General Norte* of Guayaquil, we found a mortality rate of 3.4 deaths per 10,000 person-days; and, in the *Hospital General del Sur* of Quito, the mortality rate was 1.4 per 10,000 person-days.

**Table 1** shows the characteristics of the 2,497 patients who were included in the derivation cohort. Most patients were male (63%), with a mean (standard deviation, SD) of 56.5 (16.6) years of age and a mean (SD) body mass index (BMI) of 27.0 (6.0) Kg/m$^2$. Importantly, 88.8% of patients presented a systemic inflammatory response syndrome, 33.7% had hyperglycemia, 16% had abnormal creatinine levels, and 13.6% had high blood pressure. The characteristics of the 2,565 patients in the validation cohort are shown in **S5 Table**.

**Table 1. Characteristics at the time of admission of the patients treated at the two hospitals in Quito and Guayaquil (derivation cohort).**

| Variable | Sample n = 2497 |
|---|---|
| Male sex, *n (%)* | 1569 (63) |
| Age, *mean (SD)* | 56.5 (16.6) |
| Anthropometry | |
| *Weight in Kg, mean (SD)* | 68.9 (15.4) |
| *Height in m, mean (SD)* | 1.60 (0.11) |
| *BMI in Kg/m², mean (SD)* | 27.0 (6.0) |
| Vital signs | |
| *Respiratory rate, mean (SD)* | 22.4 (4.6) |
| *Cardiac rate, mean (SD)* | 85.0 (15.0) |
| *SBP in mmHg, mean (SD)* | 119.5 (17.5) |
| *DBP in mmHg, mean (SD)* | 70.4 (11.1) |
| *Oxygen saturation, mean (SD)* | 90.9 (3.9) |
| *Body temperature in Celsius degrees, mean (SD)* | 36.8 (0.6) |
| Laboratory parameters | |
| *Glucose in mg/dL, mean (SD)* | 132.4 (74.9) |
| *Creatinine in mg/dL, mean (SD)* | 1.2 (1.7) |
| *BUN in mg/dL, mean (SD)* | 18.5 (16.1) |
| *AST in U/L, mean (SD)* | 48.7 (63.8) |
| *ALT in U/L, mean (SD)* | 52.1 (60.2) |
| *LDH en U/L, mean (SD)* | 272.4 (256.0) |
| *CPK in U/L, mean (SD)* | 180.2 (456.7) |
| *Ferritin in ng/ml, mean (SD)* | 574.1 (1115.3) |
| *C-reactive protein in mg/L, mean (SD)* | 49.2 (66.7) |
| *Procalcitonin in ng/mL, mean (SD)* | 2.59 (41.8) |
| *Arterial pH, mean (SD)* | 7.38 (0.09) |
| *$pCO_2$ in mmHg, mean (SD)* | 30.8 (9.6) |
| *White blood cell count x10³/µL, mean (SD)* | 9.27 (5.19) |
| *Lymphocytes cell count x10³/µL, media (DE)* | 1.20 (0.63) |
| *Red blood cells count x 10⁶/ µL, mean (SD)* | 4.87 (0.82) |
| *Hematocrit in percentage, mean (SD)* | 42.5 (6.5) |
| *Hemoglobin levels in g/dL, mean (SD)* | 14.5 (2.4) |
| *Platelet count x10³/µL, mean (SD)* | 275 (119) |
| *PT in seconds, mean (SD)* | 13.5 (3.2) |
| *PTT in seconds, mean (SD)* | 34 (12) |
| *D-dimer in ng/mL, mean (SD)* | 49.0 (221.5) |
| *SIRS, n (%)* | 2218 (88.8) |
| *1 point, n (%)* | 324 (13) |
| *2 points, n (%)* | 1423 (57) |
| *3 points, n (%)* | 699 (28) |
| *4 points, n (%)* | 51 (2) |
| *Clinical conditions at admission* | |
| *High blood pressure (SBP≥140 mmHg or DBP≥90 mmHg), n (%)* | 340 (13.6) |
| *Low blood pressure (SBP <90 mmHg), n (%)* | 52 (2.1) |
| *Upper than normal creatinine levels (creatinine>1.35 mg/dL if men or Creatinine>1.04 if women)* | 409 (16) |
| *Anemia, n (%)[a]* | 659 (26) |

*(Continued)*

Table 1. (Continued)

| Variable | Sample n = 2497 |
|---|---|
| *Obesity (≥30Kg/m², n (%)* | 906 (46) |

BMI = body mass index, AST = alanine transaminase, ALT = aspartate transaminase, LDH = lactic acid dehydrogenase, BUN = blood urea nitrogen, CPK = creatinine phosphokinase, PT = prothrombin time, PTT = partial thromboplastin time, SIRS = systemic inflammatory response syndrome and it means that there is, at least, two of the next criteria: Body temperature over 38 or under 36 degrees Celsius, heart rate greater than 90 beats/minute, respiratory rate greater than 20 breaths/minute or partial pressure of $CO_2$ less than 32 mmHg, leucocyte count >12 x $10^3$ or less than 4 x $10^3$ /μL or over 10% immature forms or bands.

[a] = When red blood cells low count was <3.9 for females, and when it was <4.4 for males in Guayaquil; or, when it was <4.3 for females, and when it was <5.0 for males in Quito.

We found several significant differences in the characteristics of patients who died in the hospital and those who did not (Table 2). Among those who died, there was a higher proportion of males (73% vs. 60%, p-value <0.001), the mean age was higher (66.5 vs. 53.6 years old, p-value <0.001), they had a higher respiratory rate (24.1 vs. 21.9 breaths per minute, p-value <0.001), and had a lower oxygen saturation (89.7% vs. 91.2%, p-value <0.001). Compared with survivors, deceased patients presented with significantly higher values of serum glucose (159.2 vs. 124.5 mg/dL, p-value <0.001), blood urea nitrogen (BUN) (27.0 vs. 16.1 mg/dL, p-value <0.001), lactic acid dehydrogenase (LDH) (340.2 vs. 252.1 U/L, p-value <0.001), $pCO_2$ (33.2 vs. 30.1 mmHg, p-value <0.001), and white blood cell count (12.33 vs. 8.38 × $10^3$ cells/μL, p-value <0.001); furthermore, they had lower arterial pH (7.36 vs. 7.39, p-value <0.001). In addition, deceased patients presented with a higher percentage of *(i)* systemic inflammatory response syndrome (98.1% vs. 96.1, p-value <0.001), *(ii)* hyperglycemia (55% vs. 28%, p-value <0.001), *(iii)* elevated serum creatinine (30% vs. 12%, p-value <0.001), and *(iv)* anemia (38% vs. 15%, p-value <0.001).

After multivariate analyses, we found several statistically significant and independent predictors of mortality (Table 3); male sex [(hazard ratio (HR) = 1.32, 95% confidence interval (CI): 1.03–1.69] compared to female, *per* each increase in a quartile of ages (HR = 1.44, 95% CI: 1.24–1.67) considering the younger group (17–44 years old) as the reference, presence of hypoxemia (HR = 1.40, 95% CI: 1.01–1.95), hypoglycemia and hospital hyperglycemia (HR = 1.99, 95% CI: 1.01–3.91, and HR = 1.27, 95% CI: 0.99–1.62, respectively) when compared with normoglycemia, an AST–ALT ratio >1 (HR = 1.55, 95% CI: 1.25–1.92), a C-reactive protein (CRP) level >10 mg/dL (HR = 1.49, 95% CI: 1.07–2.08), arterial pH <7.35 (HR = 1.39, 95% CI: 1.08–1.80) when compared with normal pH (7.35–7.45), and a white blood cell count >10 × $10^3$ per μL (HR = 1.76, 95% CI: 1.35–2.29). The final model is shown in detail in the S6 Table, and the comparison of accuracy and other metrics between Guayaquil and Quito can be seen in the S7 Table. Goodness of fit was assessed using the Akaike Information Criterion (AIC) and the Bayesian Information Criterion (BIC), which yielded AIC = 286.4317 and BIC = 330.3455. Chi-square tests for each variable, together with their corresponding p-values, suggest that the risks are proportional over time. The overall test p-value for the model was 0.9814, further supporting the proportional hazards assumption. These results demonstrate that the Cox regression model used in the analysis provides a good fit to the data and satisfies the proportional hazards assumption. Furthermore, when we stratified the analyses across sex categories the estimates did not modify (S8 Table). Similarly, excluding younger and older patients or those with very high hypertension or hyperglycemia did not modify the estimates (S9 Table).

**Table 2. Differences in the baseline characteristics of patients hospitalized at the two hospitals in Quito y Guayaquil, between those who did not die during hospitalization, and those who died during hospitalization (derivation cohort).**

| Variable | Did not die in hospital n = 1934 | Died in hospital n = 563 | p-value |
|---|---|---|---|
| Male sex, *n(%)* | 1158 (60) | 411 (73) | <0.001 |
| Age, *mean (SD)* | 53.6 (16.3) | 66.5 (13.3) | <0.001 |
| Anthropometry | | | |
| Weight in Kg, *mean (SD)* | 68.5 (15.0) | 70.5 (16.5) | 0.176 |
| Height in m, *mean (SD)* | 1.59 (0.10) | 1.61 (0.11) | 0.110 |
| BMI in Kg/m², *mean (SD)* | 27.0 (6.0) | 27.2 (6.0) | 0.750 |
| Vital signs at admission | | | |
| Respiratory rate, *mean (SD)* | 21.9 (4.0) | 24.1 (5.8) | <0.001 |
| Cardiac rate, *mean (SD)* | 84.8 (14.6) | 85.7 (16.3) | 0.374 |
| SBP in mmHg, *mean (SD)* | 118.6 (16.6) | 122.8 (20.0) | 0.002 |
| DBP in mmHg, *mean (SD)* | 70.6 (10.8) | 69.8 (12.0) | 0.373 |
| Oxygen saturation, *mean (SD)* | 91.2 (3.5) | 89.7 (4.9) | <0.001 |
| Body temperature in Celsius degrees, *mean (SD)* | 36.8 (0.6) | 36.7 (0.6) | 0.199 |
| Laboratory parameters | | | |
| Glucose in mg/dL, *mean (SD)* | 124.5 (68.2) | 159.2 (89.1) | <0.001 |
| Creatinine in mg/dL, *mean (SD)* | 1.1 (1.4) | 1.7 (2.4) | 0.002 |
| BUN in mg/dL, *mean (SD)* | 16.1 (13.0) | 27.0 (21.8) | <0.001 |
| AST in U/L, *mean (SD)* | 45.8 (51.6) | 58.4 (93.2) | 0.024 |
| ALT in U/L, *mean (SD)* | 52.0 (50.5) | 52.4 (85.1) | 0.931 |
| AST/ALT ratio, *mean (SD)* | | | |
| LDH in U/L *mean (SD)* | 252.1 (208.3) | 340.2 (365.2) | <0.001 |
| CPK in U/L, *mean (SD)* | 163.3 (398.0) | 236.6 (609.6) | 0.045 |
| Ferritin in ng/ml, *mean (SD)* | 550.7 (724.7) | 651.6 (1903.6) | 0.296 |
| C-reactive protein in mg/L, *mean (SD)* | 47.1 (64.2) | 56.3 (74.1) | 0.017 |
| Procalcitonin in ng/mL, *mean (SD)* | 1.5 (25.8) | 6.2 (72.9) | 0.533 |
| Arterial pH, *mean (SD)* | 7.39 (0.07) | 7.36 (0.12) | <0.001 |
| pCO₂ in mmHg, *mean (SD)* | 30.1 (8.0) | 33.2 (13.6) | <0.001 |
| White blood cell count x10³/μL, *mean (SD)* | 8.38 (4.43) | 12.33 (6.34) | <0.001 |
| Lymphocytes cell count x 10³/μL, *mean (SD)* | 1.24 (0.62) | 1.05 (0.64) | 0.005 |
| Red blood cells count x 10⁶/μL, *mean (SD)* | 4.98 (0.78) | 4.50 (0.83) | <0.001 |
| Hematocrit in percentage, *mean (SD)* | 43.1 (6.3) | 40.3 (6.6) | <0.001 |
| Hemoglobin levels in g/dL, *mean (SD)* | 14.8 (2.3) | 13.6 (2.3) | <0.001 |
| Platelet count x10³/μL, *mean (SD)* | 275 (112) | 274 (139) | 0.748 |
| PT in seconds, *mean (SD)* | 13.3 (2.3) | 14.2 (4.9) | <0.001 |
| PTT in seconds, *mean (SD)* | 33 (11) | 34 (14) | 0.396 |
| D-dimer in ng/mL, *mean (SD)* | 45.6 (201.4) | 60.4 (277.8) | 0.250 |
| SIRS, *n (%)* | 1666 (86.1) | 552 (98.1) | 0.007 |
| Specific clinical conditions at admission | | | |
| High blood pressure (see Table 1), *n (%)* | 231 (12) | 110 (20) | <0.001 |
| Low blood pressure (SBP <90 mmHg), *n (%)* | 39 (2) | 13 (2) | 0.824 |
| Upper than normal creatinine levels (see Table 1) | 239 (12) | 171 (30) | <0.001 |
| Anemia, *n (%)*[a] | 413 (21) | 246 (44) | <0.001 |

(Continued)

**Table 2.** (Continued)

| Variable | Did not die in hospital n = 1934 | Died in hospital n = 563 | p-value |
|---|---|---|---|
| *Obesity (≥30Kg/m², n (%)* | 694 (36) | 213 (38) | 0.586 |

BMI = body mass index, AST = Alanine transaminase, ALT = Aspartate transaminase, LDH = Lactic Acid Dehydrogenase, BUN = blood urea nitrogen, CPK = Creatinine phosphokinase, $pCO_2$ = partial pressure of carbon dioxide, PT = prothrombin time, PTT = partial thromboplastin time, SIRS = systemic inflammatory response syndrome and it means that there is, at least, two of the next criteria: Body temperature over 38 or under 36 degrees Celsius, heart rate greater than 90 beats/minute, respiratory rate greater than 20 breaths/minute or partial pressure of $CO_2$ less than 32 mmHg, leucocyte count >12 x $10^3$ or less than 4 x $10^3$ /μL or over 10% immature forms or bands.

[a] = When red blood cells low count was <3.9 for females, and when it was <4.4 for males in Guayaquil; or, when it was <4.3 for females, and when it was <5.0 for males in Quito

After building the score, it was named "COVID-19 in-hospital mortality score" (**Table 4**). We found an excellent discriminative ability when tested in the validation cohort (**Fig 1**), with an AUC of the ROC curve of 0.876 (95% CI: 0.822–0.930), sensitivity and specificity of 93.1% and 70.3%, respectively, and a cutoff value of ≥39 points (range, 0–86 points). The discriminative ability when tested the score in the validation cohort by hospitals and sex categories showed less discriminative capacity when the stratified analyzes were performed for the categories of the hospital and sex (**S3 Table**), when the analysis was performed in Guayaquil hospital only (AUC of the ROC curve of 0.742, 95% CI: 0.712 to 0.773) and when it was performed in females only (AUC of the ROC curve of 0.829, 95% CI: 0.671 to 0.986).

The results of the calibration analysis in the validation dataset indicated that the model demonstrated good calibration overall (**Table 5**). The chi-square tests for intercepts and slopes did not provide evidence of miscalibration (Test 1: p = 0.9321; Test 2: p = 0.7206). Furthermore, the joint test of intercepts and slopes also showed no significant evidence of miscalibration (Test 3: p = 0.9739). These findings suggest that the model's predicted event probabilities align well with the observed event probabilities in the validation dataset.

Furthermore, a cutoff score of ≥39 is considered a better cutoff over other scores because of its balance between sensitivity and specificity, along with other performance indicators. With a cut-off score of 39, a sensitivity of 93.10%, a specificity of 70.28% and a correct classification of 72.66% are achieved. In addition, the LR+ (3.1328) and LR- (0.0981) values support the usefulness of the cutoff score of 39 in identifying patients at high risk of mortality. These indicators suggest that a cut-off score of 39 provides a reasonable balance between detecting patients at risk and identifying those who are not, offering good discriminatory power.

When comparing between both hospitals, (**S10 Table**) there were differences in sensitivity, specificity, correct classification, LR+ (positive likelihood ratio) and LR- (negative likelihood ratio) between the populations of Guayaquil and Quito at each cut-off point of the COVD-19 in-hospital score tested (only show selected cut points for simplicity). It is observed that, in Guayaquil, the selected score points had higher sensitivity values, but lower specificity compared to those in Quito. In addition, the score in Guayaquil exhibited lower overall precision and more modest likelihood ratios than the score in Quito.

Also, and Based on the sensitivity and specificity, as well as the LR+ and LR- values, we can establish risk levels using the following cut-off points: low risk (score ≤ 30) with high specificity (56.18%) and LR- of 0.0000, indicating a high probability of negative results; moderate risk (score between 31 and 65) with a range of sensitivity (100% to 18.92%) and specificity (60.96% to 99.20%), suggesting a mix of positive and negative probabilities; and high risk (score > 65) with a sensitivity of 10.81%, a specificity of 99.20%, and an LR+ of 13.5675, indicating significantly higher odds of positive results compared with lower scores (**S11 Table**). These cut-off

**Table 3. Crude and adjusted associations between baseline variables at admission and in-hospital death (Cox proportional hazards models) in the derivation cohort.**

| Variable | Crude models | | Adjusted | | | |
|---|---|---|---|---|---|---|
| | HR (95% CI) | p-value | Saturated model HR (95% CI) | p-value | Parsimonious model HR (95% CI) | p-value |
| Male sex (female is the ref.) | 1.33 (1.05 to 1.69) | 0.021 | 1.24 (0.97 to 1.59) | | 1.32 (1.03 to 1.69) | 0.028 |
| Age categories | | | | | | |
| *17 to 44 years old (ref.)* | 1 | - | 1 | - | 1 | - |
| *45 to 57 years old* | 1.97 (1.1 to 3.37) | 0.015 | 1.59 (0.92 to 2.74) | 0.093 | 1.60 (0.92 to 2.78) | 0.092 |
| *58 to 68 years old* | 3.76 (2.12 to 6.65) | <0.001 | 2.52 (1.36 to 4.65) | 0.006 | 2.67 (1.48 to 4.80) | 0.003 |
| *69 to 102 years old* | 4.95 (2.92 to 8.41) | <0.001 | 2.99 (1.65 to 5.39) | 0.001 | 3.26 (1.82 to 5.84) | 0.001 |
| *p-for-trend* | 1.63 (1.43 to 1.85) | <0.001 | 1.39 (1.20 to 1.62) | <0.001 | 1.44 (1.24 to 1.67) | <0.001 |
| Vital signs | | | | | | |
| *Respiratory rate >20 per min (otherwise is the ref.)* | 1.46 (1.10 to 1.94) | 0.012 | 1.13 (0.84 to 1.52) | 0.389 | - | - |
| *Hypoxemia (no hypoxemia is the ref.)[a]* | 1.99 (1.47 to 2.68) | <0.001 | 1.41 (1.01 to 1.97) | 0.046 | 1.40 (1.01 to 1.95) | 0.043 |
| *SBP categories* | | | | | | |
| *90 to <140mmHg (ref)* | 1 | - | 1 | - | - | - |
| *<90 mm Hg* | 1.05 (0.49 to 2.25) | 0.905 | 1.00 (0.40 to 2.50) | 0.993 | - | - |
| *≥140 mmHg* | 1.38 (1.03 to 1.82) | 0.032 | 0.94 (0.34 to 2.62) | 0.904 | - | - |
| *Body temperature categories* | | | | | | |
| *36 to <37.5 Celsius degrees (ref.)* | 1 | - | 1 | - | - | - |
| *<36 Celsius degrees* | 1.11 (0.42 to 2.96) | 0.832 | 1.03 (0.37 to 2.85) | 0.956 | - | - |
| *37.5 to 37.9 Celsius degrees* | 0.85 (0.54 to 1.36) | 0.504 | 0.86 (0.53 to 1.41) | 0.539 | - | - |
| *≥38 Celsius degrees* | 0.83 (0.46 to 1.47) | 0.505 | 0.83 (0.45 to 1.54) | 0.536 | - | - |
| Laboratory parameters | | | | | | |
| *Glucose categories[b]* | | | | | | |
| *70 to ≤140 mg/dL (ref.)* | 1 | - | 1 | - | 1 | - |
| *<70 mg/dL* | 2.21 (1.19 to 4.09) | 0.012 | 1.97 (1.01 to 3.86) | 0.047 | 1.99 (1.01 to 3.91) | 0.046 |
| *>140 mg/dL* | 1.85 (1.47 to 2.32) | <0.001 | 1.21 (0.93 to 1.56) | 0.147 | 1.27 (0.99 to 1.62) | 0.056 |
| *Creatinine >1.35 mg/dL if male or >1.04 mg/dL if female (otherwise is the ref.)* | 1.83 (1.45 to 2.32) | <0.001 | 1.07 (0.79 to 1.44) | 0.656 | - | - |
| *BUN >25 mg/dL (otherwise is the ref.)* | 2.18 (1.76 to 2.69) | <0.001 | 1.11 (0.83 to 1.50) | 0.463 | - | - |
| *AST to ALT ratio >1 (otherwise is the ref.)[c]* | 1.80 (1.46 to 2.22) | <0.001 | 1.48 (1.20 to 1.82) | <0.001 | 1.55 (1.25 to 1.92) | <0.001 |
| *LDH >160 U/L(otherwise is the ref.)* | 0.89 (0.70 to 1.14) | 0.353 | 0.97 (0.69 to 1.36) | 0.831 | - | - |
| *CPK >308 U/L if male or >192 U/L if female (otherwise is the ref.)* | 1.20 (0.95 to 1.52) | 0.119 | 1.04 (0.79 to 1.36) | 0.782 | - | - |
| *C-reactive protein >10 mg/dL (otherwise is the ref.)* | 1.68 (1.21 to 2.33) | 0.003 | 1.46 (1.03 to 2.08) | 0.037 | 1.49 (1.07 to 2.08) | 0.022 |
| *Arterial pH categories* | | | | | | |
| *7.35 to 7.45 (ref.)* | 1 | - | 1 | - | 1 | - |
| *<7.35* | 1.84 (1.41 to 2.40) | <0.001 | 1.24 (0.94 to 1.63) | 0.119 | 1.39 (1.08 to 1.80) | 0.014 |
| *>7.45* | 1.31 (1.03 to 1.67) | 0.028 | 1.11 (0.85 to 1.44) | 0.452 | 1.08 (0.84 to 1.39) | 0.553 |
| *$pCO_2$ categories* | | | | | | |
| *35 to <46 mmHg* | 1 | - | 1 | - | - | - |
| *<35 mmHg* | 0.90 (0.69 to 1.19) | 0.460 | 0.96 (0.73 to 1.26) | 0.752 | - | - |
| *≥46 mmHg* | 1.78 (1.24 to 2.57) | 0.002 | 1.20 (0.82 to 1.75) | 0.343 | - | - |
| *White blood cell count* | | | | | | |
| *>10 x10³ per μL (otherwise is the ref.)* | 2.32 (1.85 to 2.92) | <0.001 | 1.71 (1.35 to 2.18) | <0.001 | 1.76 (1.35 to 2.29) | <0.001 |
| *Lymphocytes cell count* | | | | | | |
| *1.1 to 3.2 x10³ per μL (ref.)* | 1 | - | 1 | - | - | - |

*(Continued)*

**Table 3.** (Continued)

| Variable | Crude models | | Adjusted | | | |
|---|---|---|---|---|---|---|
| | HR (95% CI) | p-value | Saturated model HR (95% CI) | p-value | Parsimonious model HR (95% CI) | p-value |
| <1 x10³ per μL | 1.39 (1.12 to 1.72) | 0.003 | 1.09 (0.85 to 1.39) | 0.484 | - | - |
| >3.2 x10³ per μL | 1.43 (0.34 to 6.0) | 0.622 | 1.43 (0.317 to 6.41) | 0.632 | - | - |
| *Red blood cells count categories[d]* | | | | | | |
| *Normal count (ref.)* | 1 | - | 1 | - | - | - |
| *Low count* | 1.66 (1.33 to 2.08) | <0.001 | 1.08 (0.81 to 1.43) | 0.601 | - | - |
| *High count* | 0.64 (0.31 to 1.35) | 0.243 | 0.69 (0.34 to 1.41) | 0.303 | - | - |
| *Platelet count categories* | | | | | | |
| *150 to 450 x10³ per μL (ref.)* | 1 | - | 1 | - | | |
| <150 x10³ per μL | 1.31 (0.87 to 1.97) | 0.187 | 1.30 (0.87 to 1.96) | 0.189 | - | - |
| >450 x10³ per μL | 0.85 (0.56 to 1.29) | 0.448 | 0.74 (0.49 to 1.12) | 0.155 | - | - |
| *PT >14.5 sec (otherwise is the ref.)* | 1.49 (1.07 to 2.06) | <0.001 | 1.14 (0.85 to 1.52) | 0.359 | - | - |
| *D-dimer (otherwise is the ref.)* | 0.93 (0.64 to 1.37) | 0.727 | 1.13 (0.72 to 1.78) | 0.571 | - | - |
| *Specific clinical conditions at admission* | | | | | | |
| *SIRS (otherwise is the ref.)* | 1.37 (0.86 to 2.18) | 0.174 | 1.13 (0.71 to 1.81) | 0.591 | - | - |
| *High blood pressure (otherwise is the ref.)* | 1.32 (0.99 to 1.78) | 0.056 | 1.20 (0.38 to 3.76) | 0.744 | - | - |

HR = hazard ratios, 95% CI = 95% confidence interval, BMI = body mass index, AST = Alanine transaminase, ALT = Aspartate transaminase, LDH = Lactic Acid Dehydrogenase, BUN = blood urea nitrogen, CPK = Creatinine phosphokinase, pCO2 = partial pressure of carbon dioxide, PT = prothrombin time, PTT = partial thromboplastin time, SIRS = systemic inflammatory response syndrome and it means that there is, at least, two of the next criteria: Body temperature over 38 or under 36 degrees Celsius, heart rate greater than 90 beats/minute, respiratory rate greater than 20 breaths/minute or partial pressure of CO2 less than 32 mmHg, leucocyte count >12 x 103 or less than 4 x 103 /μL or over 10% immature forms or bands.

[a] = Hyperglycemia in hospitalized patients is defined as blood glucose levels >140 mg/dL [30].

[b] = hypoxemia was defined when oxygen saturation at admission was <95% in Guayaquil (0 meters above the sea level) and <92% in Quito (2885 meters above the sea level).

[c] = A normal AST:ALT ratio should be <1 [31].

[d] = Red blood cells normal count (x 10¹²) was defined when it was ≥3.9 to <5.5 for females, and when it was ≥4.4 to <6.0 for males in Guayaquil; or, when it was ≥4.3 to <5.7 for females, and when it was ≥5.0 to <6.4 for males in Quito. Red blood cells low count was defined when it was <3.9 for females, and when it was <4.4 for males in Guayaquil; or, when it was <4.3 for females, and when it was <5.0 for males in Quito. Red blood cells high count was defined when it was ≥5.5 for females, and when it was ≥6.0 for males in Guayaquil; or, when it was ≥5.7 for females, and when it was ≥6.4 for males in Quito [32].

points help classify people into different levels of risk based on the balance between sensitivity and specificity. The Kaplan–Meier curves were statistically significant (Log-Rank test, p-value <0.001) when comparing the survival between intermediate-risk, high-risk, and very high-risk groups in both cohorts (**Fig 2**).

When comparing between both hospitals, (**S12 Table**) there are differences in sensitivity, specificity, correct classification, LR+ (positive likelihood ratio) and LR- (negative likelihood ratio) between the populations of Guayaquil and Quito at each cut-off point of the COVD-19 in-hospital score tested (only show selected cut points for simplicity). It is observed that, in Guayaquil, the selected score points had higher sensitivity values, but lower specificity compared to those in Quito. In addition, the score in Guayaquil exhibited lower overall precision and more modest likelihood ratios than the score in Quito.

## Discussion

Comparing patients who died with those who survived presented a significantly higher frequency of male sex, older age, hypoxemia, hypoglycemia, or hospital hyperglycemia (a hospital

**Table 4. The COVID-19 in-hospital mortality score.**

| Risk factor | Addition to risk score | Risk score |
|---|---|---|
| Male sex | 6 | |
| Age categories | | |
| 45 to 57 years old | 9 | |
| 58 to 68 years old | 20 | |
| 69 to 102 years old | 24 | |
| Hypoxemia[a] | 7 | |
| Laboratory parameters | | |
| Glucose categories | | |
| <70 mg/dL | 14 | |
| >140 mg/dL | 5 | |
| AST to ALT ratio >1 | 9 | |
| C-reactive protein >10 mg/dL | 8 | |
| Arterial pH categories | | |
| <7.35 | 7 | |
| >7.45 | 2 | |
| White blood cell count categories | | |
| >10 x10³ /μL | 9 | |
| | Total Risk Score | |

Interpretation: 0 to 30 points = moderate risk, 31 to 65 points = high risk, 66 to 86 points = very high risk.

[a] = hypoxemia was defined when oxygen saturation at admission was <95% in Guayaquil (0 meters above the sea level) and <92% in Quito (2885 meters above the sea level).

*AST = aspartate aminotransferase; ALT = Alanine transaminase

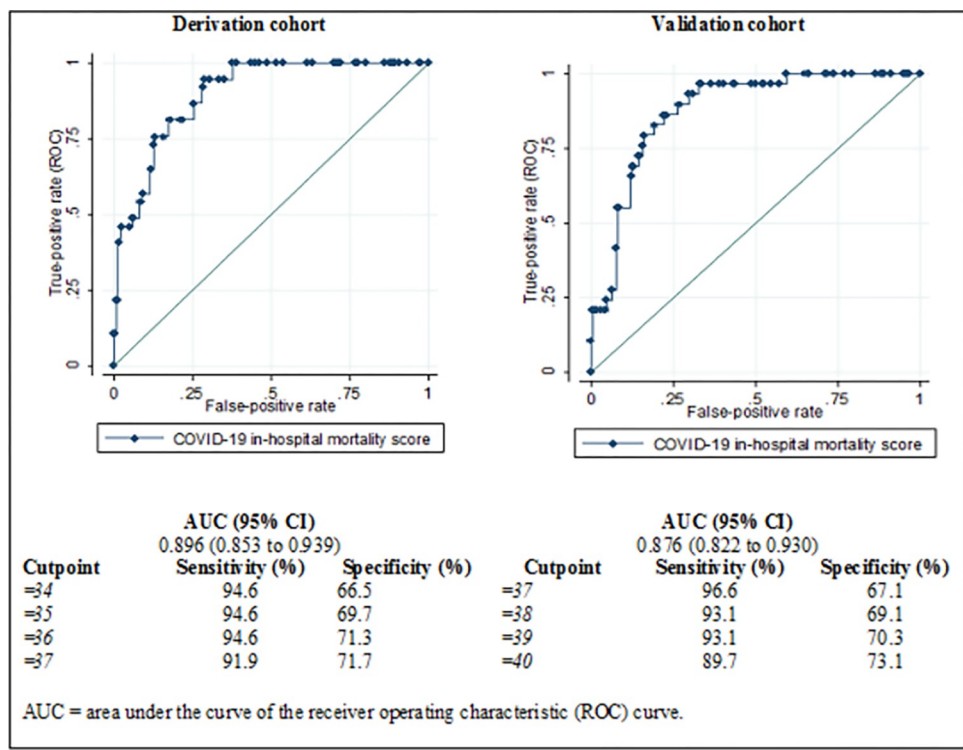

**Fig 1. Area under the curve of the receiver operating characteristic (ROC) curve of the COVID-19 in-hospital mortality score in both, the derivation and validation cohorts.**

**Table 5. Results of the calibration analysis in the validation dataset.** The intercept test, slope test, joint test, and interaction test were performed to evaluate the model's calibration.

| Test | Chi-square | p-value |
|---|---|---|
| Intercept test | 0.14 | 0.9321 |
| Slope test | 0.13 | 0.7206 |
| Joint test | 0.22 | 0.9739 |
| Interaction test | 0.03 | 0.8729 |

glucose level >140 mg/dL) [30], AST–ALT ratio >1, elevated CRP, and altered arterial pH. These factors were used to construct a prognostic clinical scoring system, the "COVID-19 in-hospital mortality score", with the excellent ability to discriminate the risk of mortality in patients hospitalized for COVID-19.

In the presented study, it was observed that the "COVID-19 in-hospital mortality score" demonstrated greater utility in predicting mortality in unvaccinated patients and those residing in higher altitude areas. It is important to note that this finding may be related to specific biological adaptations and environmental factors in these populations [53, 54]. However, further research in diverse populations and contexts is required to confirm and better understand these findings.

Higher altitude populations have been shown to exhibit a lower rate of COVID-19 related mortality in some studies [33]. This could be due to biological adaptations to hypoxia, which may play a role in the body's response to the virus. Additionally, specific environmental factors of high-altitude areas could influence the spread and severity of the disease [34]. On the other hand, in unvaccinated populations, the lack of immunity acquired from vaccination might lead to more severe disease outcomes, which could explain the higher utility of the score in these populations.

Nevertheless, it was found that the score exhibited lower discriminative ability when analyzed in patients from Guayaquil's hospital and in women [S2 Table]. This suggests that the score might be less effective in certain population groups and contexts, emphasizing the importance of validating and adapting the score to different populations and settings.

In conclusion, although the "COVID-19 in-hospital mortality score" appears to be particularly useful in unvaccinated patients and those residing in higher altitude areas, further

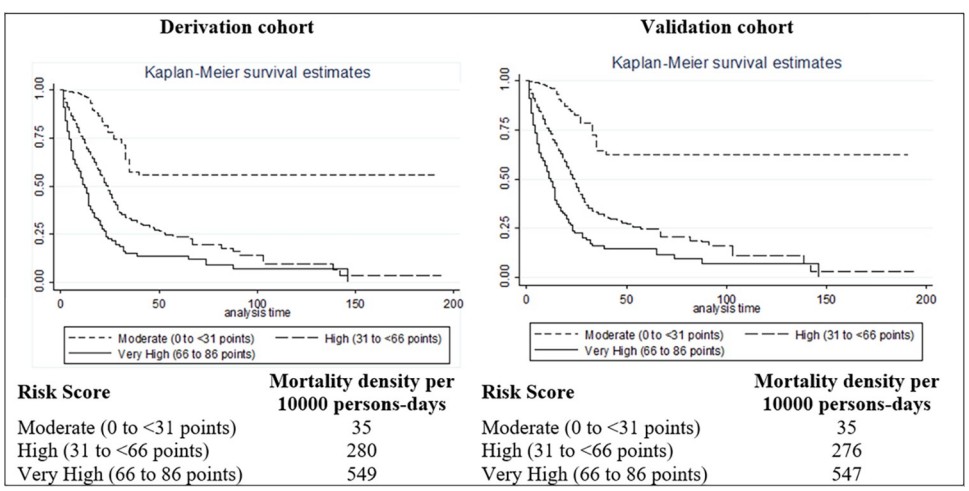

**Fig 2. Kaplan–Meier survival curves for the derivation and the validation cohort, according to the prognostic classification categories of the COVID-19 in-hospital mortality score.**

research is necessary to validate and adapt this score to different populations and contexts. This would enhance its applicability and utility in clinical practice and healthcare resource planning. Several studies have proposed mortality scoring systems in patients hospitalized for COVID-19 [35]. The most common predictors of in-hospital death used in these scores are *(i)* male sex [36], *(ii)* older age [37], *(iii)* presence of comorbidities (such as high blood pressure and diabetes) [38, 39], *(iv)* alteration of vital signs (such as hypoxemia) [40], and *(v)* obesity [41]. Moreover, important laboratory parameters predicting a worse prognosis included: increased concentrations of BUN [42], creatinine [43], serum glucose [44], AST [45], increased AST–ALT ratio [46], LDH [47], creatinine phosphokinase (CPK) [48], CRP [49], procalcitonin [50], ferritin [51, 52], altered blood count parameters, such as leucocytosis, leukopenia [53], lymphocytosis [54], lymphopenia [53], altered neutrophil and lymphocyte index [55], thrombocytopenia [56], and thrombocytosis [55], as well as, alteration of the prothrombin time [57], and increase in the concentration of D-dimer [52]. Present results are consistent with these investigations, supporting the feasibility for the clinical use of the "COVID-19 in-hospital mortality score". As a systematic review showed, scoring systems for predicting severe COVID-19 showed an AUC from 0.52 to 0.98 [58], with most articles from China, Europe, and North America, where COVID- 19 hit the hardest. In the same review, despite prediction models reporting validation, most studies have a significant risk of bias, likely due to limited transparency on the predictor and outcome assessment and a limited number of samples. In that sense, and despite our retrospective approach, we overcame the small sample size of other studies, the outcome was clearly defined, and overfitting was not a limitation of the current study. Nevertheless, when we circumscribed the calculation of the AUC to female hospitalized patients and to patients of the Guayaquil hospital the AUC was reduced.

These findings could be explained by several factors: (i) the score could be less effective in predicting in-hospital death at the sea level, maybe the lower COVID-19 related deaths in altitude were the results from a biological adaptation to hypoxemia, due to environmental factors [34], or related to the different contexts between both cities/altitudes [59], (ii) the score could be less effective in predicting in-hospital death for females maybe because of the lack of information regarding comorbidities which are, generally, more frequent among females [60], and (iii) the lack of a similar representativeness of Guayaquil and female population by comparison with their counterparts (patients from Quito and male), when building the score. In summary, the score seems to work best for hospitalized men in Quito. Moreover, we strongly recommend the external validation of the "COVID-19 in-hospital mortality score" in independent populations to assess its generalizability and applicability across different settings. Future validation studies involving diverse patient cohorts will provide valuable insights into the performance and robustness of the scoring system, ensuring its reliability and usefulness in clinical practice.

Regarding the clinical course and risk factors for mortality of adult patients with COVID-19 in the studied population, we found that mortality rates between both hospitals were different (338 vs. 144 per 10,000 person-days). As other studies have found, we believe that context differences in both cities and the health system's response adaptability factors could contribute to such differences in the mortality rate. We speculate that two main factors could explain the differences in mortality: *(i)* Guayaquil had a more severe impact from the pandemic at the very beginning, collapsing the health system and creating the impossibility of organizing the system in the face of overcrowded health services; and, *(ii)* as several studies have confirmed, context factors such as altitude [34] could improve survival in Quito (2850 m) by comparison with Guayaquil (4 m). In that sense, we recognize that we did not have enough information about the differences in the health system organization and context information. In that regard, we recommend further investigation to confirm such context determinants on COVID-19 mortality.

Moreover, the calibration analysis of the Cox proportional hazards model in the validation dataset demonstrated its ability to accurately estimate event probabilities. This indicates that the model can be considered reliable for predicting outcomes in similar population; Moreover, the tests of calibration, including the interaction between the slope and times, did not show significant differences, supporting the model's adequacy in terms of calibration over time.

The scoring system developed in the present study has potential implications in clinical settings and in the field of public health. Specifically, we believe that this score could allow clinicians to make timely decisions to stratify the risk of patients hospitalized with COVID-19 [61]. In the field of public health, the use of the "COVID-19 in-hospital mortality score" could contribute to a rational allocation of resources for health care [62], since risk bands could be identified early with the timely allocation of resources and clinical management [63].

We believe that our study has some strengths. It was carried out in a real scenario, with an important sample of patients from the two main Ecuadorian cities, which also have very different geographic, social, economic, and climatic characteristics, allowing estimates with proper external validity. As a retrospective cohort study, it is possible to estimate mortality risk factors [64] with potential use for scoring systems.

We recognize some limitations in our study; however, these are not significant sources of bias. Probably the most important limitation is the lack of information on a majority of the variables for the patients included in the sample. Although the percentage of missing values was high, the use of multiple imputation techniques, using chained equations, have resulted in realistic estimates with standard errors very close to reality and that have already been used for studies like ours [65].

Another possible limitation to the present study is the lack of information concerning other possible health predictors, such as the need for mechanical ventilation, use of intensive care therapy, and use of inotropic drugs. Unfortunately, the database administration process prevented us from carrying out these analyses because of the urgency in responding to other pandemic health necessities. Currently, it is impossible to link the anonymized information with that from the medical records. We recognize the possible source of residual confusion in the lack of information regarding ethnicity, deprivation, specific symptoms and clinical signs, sociodemographic factors, and specific comorbidities that have been shown to be possible predictors of poor health outcomes in patients with COVID-19. Nevertheless, according to a systematic review [58] that found nine studies reporting prediction models that were rated as low risk of bias and low concerns for applicability, several scoring systems that allow proper prognosis in patients with COVID-19 do not always include these variables. Therefore, we believe that this is a source of residual confounding that is unlikely to bias our estimates substantially. A final limitation is the lack of validation of the score in a different population since the internal and external validation was carried out in the same group of patients. In that sense, we encourage a process of validation of our scoring system in other contexts.

## Conclusion

Male sex, increasing age, hypoxemia, hypoglycemia or hospital hyperglycemia, AST–ALT ratio >1, elevated CRP, altered arterial pH, and leucocytosis were factors significantly associated with higher mortality in patients hospitalized with COVID-19. These factors were included to construct a prognostic clinical score system with the ability to determine the risk of mortality in patients hospitalized for COVID-19. Based on the statistical significance of the Cox regression model, its discriminatory ability (AUC-ROC), and calibration, the results suggest that the developed model is a valuable tool for predicting mortality in hospitalized COVID-19 patients. The model demonstrates robust statistical performance, with significant

associations between the predictor variables and mortality outcomes. Furthermore, the model exhibits good discrimination, as evidenced by a high AUC-ROC value. The calibration analysis confirms the model's ability to accurately estimate mortality probabilities. These findings collectively indicate the potential clinical utility of the model in identifying individuals at higher risk of mortality and informing appropriate interventions.

## Supporting information

**S1 Data. Imputed dataset.**
(XLSX)

**S1 Table. Prediction model development and validation checklist (adapted from (1))—for illustrative purposes only.**
(DOCX)

**S2 Table. Comparison of characteristics between imputed and non-imputed data sets (derivation cohort).**
(DOCX)

**S3 Table. Comparison of the discriminatory power of the score between hospitals and between sex categories, calculating the area under the ROC curve in the validation cohort.**
(DOCX)

**S4 Table. Summary of the Cox model calibration results for Quito.**
(DOCX)

**S5 Table. Characteristics at the time of admission of the patients treated at the two hospitals in Quito and Guayaquil (validation cohort).**
(DOCX)

**S6 Table. Cox regression modelling with multiple imputations was applied.**
(DOCX)

**S7 Table. Comparison of accuracy and other metrics between Guayaquil an Quito.**
(DOCX)

**S8 Table. Crude and adjusted associations between baseline variables at admission and in-hospital death, according to our parsimonious model on Table 3 of the main text (Cox proportional hazards model) stratified by sex (validation cohort).**
(DOCX)

**S9 Table. Crude and adjusted associations between baseline variables at admission and in-hospital death, according to our parsimonious model on Table 3 of the main text (Cox proportional hazards model) excluding: (i) patients in both extreme of age quartiles, (ii) patients with ≥180 mmHg of systolic blood pressure or ≥90 mmHg of diastolic blood pressure; and (iii) patients with ≥200 mg/dL of serum glucose (validation cohort).**
(DOCX)

**S10 Table. Comparative detailed report of sensitivity and specificity between Guayaquil and Quito.**
(DOCX)

**S11 Table. Sensitivity, specificity, and classification accuracy of the proposed risk classification cutpoints.**
(DOCX)

**S12 Table. Comparative detailed report of sensitivity and specificity between Guayaquil and Quito.**
(DOCX)

## Acknowledgments

Authors thank the contributions from the personnel in charge of anonymizing data from *Hospital Los Ceibos*, Guayaquil, and *Hospital General del Sur de Quito—IESS*, Quito.

## Author Contributions

**Conceptualization:** Iván Dueñas-Espín, María Echeverría-Mora, Camila Montenegro-Fárez.

**Data curation:** Iván Dueñas-Espín, María Echeverría-Mora, Camila Montenegro-Fárez.

**Formal analysis:** Iván Dueñas-Espín.

**Funding acquisition:** Iván Dueñas-Espín, María Echeverría-Mora, Camila Montenegro-Fárez.

**Investigation:** Iván Dueñas-Espín, María Echeverría-Mora, Camila Montenegro-Fárez, Manuel Baldeón, Luis Chantong Villacres, Hugo Espejo Cárdenas, Marco Fornasini, Miguel Ochoa Andrade, Carlos Solís.

**Methodology:** Iván Dueñas-Espín, María Echeverría-Mora, Camila Montenegro-Fárez.

**Project administration:** Iván Dueñas-Espín.

**Validation:** Iván Dueñas-Espín, María Echeverría-Mora, Camila Montenegro-Fárez, Manuel Baldeón, Luis Chantong Villacres, Hugo Espejo Cárdenas, Marco Fornasini, Miguel Ochoa Andrade, Carlos Solís.

**Writing – original draft:** Iván Dueñas-Espín, María Echeverría-Mora, Camila Montenegro-Fárez.

**Writing – review & editing:** Iván Dueñas-Espín, María Echeverría-Mora, Camila Montenegro-Fárez, Manuel Baldeón, Luis Chantong Villacres, Hugo Espejo Cárdenas, Marco Fornasini, Miguel Ochoa Andrade, Carlos Solís.

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
