## [Decision Letter · Decision Letter 0]

5 Dec 2022

PONE-D-22-16810Development and validation of a scoring system to predict mortality in patients hospitalized with COVID-19: a retrospective cohort study in two large hospitals in EcuadorPLOS ONE

Dear Dr. Dueñas-Espín,

Thank you for submitting your manuscript to PLOS ONE. After careful consideration, we feel that it has merit but does not fully meet PLOS ONE’s publication criteria as it currently stands. Therefore, we invite you to submit a revised version of the manuscript that addresses the points raised during the review process.

Thanks for your patience, it has been difficult to find reviewers and as you can see on the comments there was quite a lot of discrepancy amongst them. As two of them were quite positive we would like you to address all the comments, specially those that are associated with the reviewer that proposed to reject the manuscript. The issues raised by that reviewer are important and we would like to see a revised version of the manuscript before we can further assess its suitability.

We look forward to receiving your revised manuscript.

Kind regards,

Monica Cartelle Gestal, PhD

Academic Editor

PLOS ONE

https://journals.plos.org/plosone/s/fileid=ba62/PLOSOne_formatting_sample_title_authors_affiliations.pdf.

Reviewers' comments:

Reviewer's Responses to Questions

**Comments to the Author**

1. Is the manuscript technically sound, and do the data support the conclusions?

Reviewer #1: Yes

Reviewer #2: Partly

Reviewer #3: Yes

2. Has the statistical analysis been performed appropriately and rigorously? 

Reviewer #1: Yes

Reviewer #2: Yes

Reviewer #3: Yes

3. Have the authors made all data underlying the findings in their manuscript fully available?

Reviewer #1: No

Reviewer #2: Yes

Reviewer #3: Yes

4. Is the manuscript presented in an intelligible fashion and written in standard English?

Reviewer #1: Yes

Reviewer #2: Yes

Reviewer #3: Yes

5. Review Comments to the Author

Reviewer #1: This document is considered to be of good quality and that it provides a lot of relevant information to understand and, above all, consider certain aspects in the care of COVID patients. Some small observations are presented in order to improve the contribution of the proposed article.

Reviewer #2: In their manuscript, the authors build a risk scoring system for mortality during hospitalization for COVID-19. While the paper is of interest, there are some major technical issues that have to be addressed.

Major comments:

Not having the ability to include the other clinical information- mechanical ventilation and drug/antibody usage in the risk scoring system is a major limitation and has to be included as those are two of the most common interventions and have shown to have predictive value in recovery.

Minor comments:

Introduction- numbers for wordwide coronavirus statistics need to be updated, maybe mention which wave of variant caused the most hospitalizations

Omicron is mentioned but 2022 supporting omicron’s severity is not included

Unavailability of testing- is this in reference to early in the pandemic? This statement may need further clarification

Percentage of booster uptake currently?

Conclusions-Elevated RT-PCR- what does that even mean? Higher viral load? Only can make that correlation if PCR is done with standard curve, unclear in methods

Reviewer #3: This manuscript by Dueñas-Espín et al described a retrospective study. One major limitation of the study is a large (28.6%) percentage of missing data. However the authors are transparent about the limitation, have used randomized methods to impute the missing data and have performed the analysis properly.

The manuscript is well-written. Even though findings are not surprinsing, it is consistent with what is expected from clinical studies of COVID-19 patients.

Despite the limitations, the authors should be commneded for such a large study that includes thousands of patients.

I think the authors do need to make some changes to make the manuscript better. Introduction section of the manuscript needs to be updated throughly with more recent statistics and appropirate references.

In the abstract, include the abbreviation CRP and change "elevated PCR" to "elevated CRP".

Result section, "2,565 patients are shown in S2 Table" should be "2,565 patients in the validation cohort are shown in S2 Table"

6. PLOS authors have the option to publish the peer review history of their article (what does this mean?). If published, this will include your full peer review and any attached files.

Reviewer #1: No

Reviewer #2: No

Reviewer #3: No

---

## [Author Response · Author response to Decision Letter 0]

24 Feb 2023

Rebuttal letter to reviewers’ comments to authors

PLOS ONE

Manuscript ID: PONE-D-22-16810

Title: Development and validation of a scoring system to predict mortality in patients hospitalized with COVID-19: a retrospective cohort study in two large hospitals in Ecuador

Authors: Iván Dueñas-Espín, María Echeverría-Mora, Camila Montenegro-Fárez, Manuel Baldeón, Luis Chantong Villacres, Hugo Espejo Cárdenas, Marco Fornasini, Miguel Ochoa Andrade, Carlos Solís.

Decision: Revision required

Article Type: original article.

Corresponding Author: Iván Dueñas-Espín

We thank the Editor and Reviewers for their insightful comments that have been helpful to improve the manuscript. These comments are very much appreciated. This retrospective cohort study generated a system scoring model to predict the mortality of hospitalized COVID-19 patients at two large hospitals in Ecuador. This model found a higher COVID-19 mortality risk for males, patients of increasing age, patients with hypoxia, hypoglycemia, hyperglycemia, an AST to ALT ratio >1, elevated C-reactive protein, altered arterial pH, and leucocytosis. We believe that our study makes a significant contribution to the literature because the findings of this study are consistent with previous research, but some of the limitations of the previous studies were overcome in our study. Further, we believe that this paper will be of interest to the readership of your journal because COVID-19 is still a major problem globally, and this model could be used by physicians to treat patients at higher risk of mortality from COVID-19, therefore saving lives. We hope that our answers satisfy the questions and comments from the Editorial team and the reviewers.

 

Dear Dr. Dueñas-Espín,

Thank you for submitting your manuscript to PLOS ONE. After careful consideration, we feel that it has merit but does not fully meet PLOS ONE’s publication criteria as it currently stands. Therefore, we invite you to submit a revised version of the manuscript that addresses the points raised during the review process.

Thanks for your patience, it has been difficult to find reviewers and as you can see on the comments there was quite a lot of discrepancy amongst them. As two of them were quite positive we would like you to address all the comments, especially those that are associated with the reviewer that proposed to reject the manuscript. The issues raised by that reviewer are important and we would like to see a revised version of the manuscript before we can further assess its suitability.

ANSWER:

Thanks a lot for the insightful comments that reviewers and editorial team have done to our manuscript. We believe that, after the revision, we have answered properly all the comments in order to achieve suitability for being published in your prestigious journal.

 

JOURNAL REQUIREMENTS:

https://journals.plos.org/plosone/s/fileid=ba62/PLOSOne_formatting_sample_title_authors_affiliations.pdf.

ANSWER:

We have reviewed the guidelines and changed the manuscript accordingly. Please see marked and unmarked copies of the manuscript. 

ANSWER:

Thank you very much. After a brief discussion with the authors of the manuscript we decided to make the data public for proper replication of our results. We have changed the “Data Availability” statement in this way: “All relevant data are within the manuscript and its Supporting Information files”. Therefore, we share the database as a supporting information file “S1 Data” as a “.csv” file. In that regard, we have changed the Data Availability statement as follows: 

“Data Availability Statement: All relevant data are within the manuscript and its Supporting information files.”

This new statement has been mentioned in the new covering letter. 

COMMENTS TO THE AUTHOR

1. Is the manuscript technically sound, and do the data support the conclusions?

Reviewer #1: Yes

Reviewer #2: Partly

Reviewer #3: Yes

ANSWER:

Thank you for your positive feedback, we have clarified and completed several parts of the main text and supplementary material to improve the coherence along the whole manuscript.

2. Has the statistical analysis been performed appropriately and rigorously?

Reviewer #1: Yes

Reviewer #2: Yes

Reviewer #3: Yes

ANSWER:

Thank you for your positive feedback.

3. Have the authors made all data underlying the findings in their manuscript fully available?

Reviewer #1: No

Reviewer #2: Yes

Reviewer #3: Yes

ANSWER:

Thank you for your positive feedback. We have modified the Data Availability Statement. Now, we share the database as a supporting information file “S1 Data” as a “.csv” file.

4. Is the manuscript presented in an intelligible fashion and written in standard English?

Reviewer #1: Yes

Reviewer #2: Yes

Reviewer #3: Yes

ANSWER:

Thank you for your positive feedback.

 

5. Review Comments to the Author

ANSWERS TO REVIEWERS

ANSWERS TO REVIEWER #1:

Reviewer #1: This document is considered to be of good quality and that it provides a lot of relevant information to understand and, above all, consider certain aspects in the care of COVID patients. Some small observations are presented in order to improve the contribution of the proposed article.

ANSWER:

Thanks a lot for your important feedback. We have carefully read your comments and we have changed the manuscript according to satisfy them.

“At present, despite the two-dose vaccination prevalence being close to 77.87 % (one of the highest in the region) [12,16], there are difficulties reaching populations where vaccination coverage is complex (i.e., rural areas population). In that sense, a notable increase in cases has been reported during January 2022 [7].” 

It is unlikely that the only and main cause of the increase in cases is due to vaccination coverage, it would be important to analyze the possible impact of the mobilization of the population for vacation periods or other festivities that impacted the increase in cases, relaxation of protection measures, distancing in the population, restart of economic activities, increase in capacity in establishments, etc.

ANSWER:

Tanks a lot for your important observation. We have reviewed literature and concluded the same as you. The factors which affect negatively for a further dissemination of the disease goes far away than the simple lack of vaccination coverage. In that regard we have added next text in the “Introduction” section, lines 96 to 111, page 6 and 7, as follows: 

“Among the different possible explanations, the following aspects are the ones that could mainly explain it: (i) vaccines are not 100% effective, (ii) unvaccinated people could be fueling transmission, (iii) spreading the disease by vaccinated cases, (iv) people that are moving around more to restart economic activities or during vacation periods and festivities, (v) the relaxation of protection measures like distancing measures or the increase in capacity in establishments; and (vi) the current global dominance of the SARS-CoV-2 Omicron (BA.1 or B.1.1.529) and several subvariants that are appearing more frequently, with a higher transmissibility, even in vaccinated individuals (1)…”

References:

1. Khandia, R., Singhal, S., Alqahtani, T., Kamal, M. A., El-Shall, N. A., Nainu, F., Desingu, P. A., & Dhama, K. (2022). Emergence of SARS-CoV-2 Omicron (B.1.1.529) variant, salient features, high global health concerns and strategies to counter it amid ongoing COVID-19 pandemic. In Environmental Research (Vol. 209). Academic Press Inc. https://doi.org/10.1016/j.envres.2022.112816.

2. Hyams, C., Challen, R., Marlow, R., Nguyen, J., Begier, E., Southern, J., King, J., Morley, A., Kinney, J., Clout, M., Oliver, J., Gray, S., Ellsbury, G., Maskell, N., Jodar, L., Gessner, B., McLaughlin, J., Danon, L., Finn, A., … Szasz-Benczur, Z. (2023). Severity of Omicron (B.1.1.529) and Delta (B.1.617.2) SARS-CoV-2 infection among hospitalised adults: A prospective cohort study in Bristol, United Kingdom. The Lancet Regional Health - Europe, 25, 1–11. https://doi.org/10.1016/j.lanepe.2022.100556. 

The groups analyzed should also be considered with the issue of the height of the sea level, considering that one city is above 2,800 meters and the other city is at sea level. This could be a factor that should be taken into account when analysing and comparing the results. It would be interesting to have the analysis by cities. Some parameters analyzed could have an impact for the simple reason of the altitude in which they are living.

ANSWER:

We thank the reviewer. Your comment has been of great value to enrich the content of our manuscript. We have to mention that, indeed, it was planned to perform the analyses of the discriminatory power of the score between hospitals and between sex categories, calculating the Area under the ROC curve in the validation cohort. We kindly apologize for the omission. In that regard, we show the results in the new S2 Table. “Comparison of the discriminatory power of the score between hospitals and between sex categories, calculating the Area under the ROC curve in the validation cohort.” The order of the other supplementary tables were changed according to the insertion of the new table. In that sense, we modified the text at line 214 of page 11 of the “Material and Methods” section adding:

“We compared the discriminatory power of the score between hospitals and between sex categories, calculating the AUC in the two hospitals and both sex categories (S2 Table)”.

Furthermore, in the “Results” section (first paragraph of 19th page), in which added a text as follows:

“The discriminative ability when tested the score in the validation cohort by hospitals and sex categories showed less discriminative capacity when the stratified analyses were performed for the categories of the hospital and sex variables (S2 Table), especially when the analysis was performed in Guayaquil hospital (AUC of the ROC curve of 0.742, 95% CI: 0.712 to 0.773) and when it was performed in females (AUC of the ROC curve of 0.829, 95% CI: 0.671 to 0.986).”

Finally, at the light of those findings, we added a text in the “Discussion” section, in the last paragraph of the page 20, as follows:

“Nevertheless, when we circumscribed the calculation of the AUC to female hospitalized patients and to patients of the Guayaquil hospital, the AUC was reduced. These findings could be explained by several factors: (i) the score could be less effective in predicting in-hospital death at the sea level, maybe the lower COVID-19 related deaths in altitude were the results from a biological adaptation to hypoxia, due to environmental factors, (3) or related to the different contexts between both cities/altitudes (4), (ii) the score could be less effective in predicting in-hospital death for females maybe because of the lack of information regarding comorbidities which are, generally, more frequent among females (5), and (iii) the lack of a similar representativeness of Guayaquil and female population by comparison with their counterparts (patients from Quito and male), when building the score. In summary, the score seems to work best for hospitalized men in Quito.”

References:

3. Simbaña-Rivera K, Jaramillo PRM, Silva JVV, Gómez-Barreno L, Campoverde ABV, Novillo Cevallos JF, et al. High-altitude is associated with better short-term survival in critically ill COVID-19 patients admitted to the ICU. PLoS One. 2022;17(3):e0262423. 

4. Abbasi BA, Chanana N, Palmo T, Pasha Q. Disparities in COVID-19 incidence and fatality rates at high-altitude. 2023;2:1–16. 

5. Gu H, Wang C, Zhang X, Jiang Y, Li H, Meng X. Ten-Year Trends in Sex Differences in Cardiovascular Risk Factors, In-Hospital Management, and Outcomes of Ischemic Stroke in China: Analyses of a Nationwide Serial Cross-Sectional Survey from 2005 to 2015. Int J Stroke. 2023;Feb 8:1747(Epub ahead of print):2023. 

 

ANSWERS TO REVIEWER 2

Reviewer #2: In their manuscript, the authors build a risk scoring system for mortality during hospitalization for COVID-19. While the paper is of interest, there are some major technical issues that have to be addressed.

Major comments:

Not having the ability to include the other clinical information- mechanical ventilation and drug/antibody usage in the risk scoring system is a major limitation and has to be included as those are two of the most common interventions and have shown to have predictive value in recovery.

ANSWER:

Thanks a lot for your accurate observation, it has been very useful for improving the manuscript. Unfortunately, the database did not include information about other clinical information as the use of mechanical ventilation nor the use of monoclonal antibodies; therefore, it is impossible to test those variables to assess if the discriminatory ability of the score improves. In any case, we looked for other possibilities to try to explore the potential effect of these important variables. 

Regarding the mechanical ventilation usage, evidence shows that the use of mechanical ventilation is very associated with a high mortality in patients with COVID-19; nevertheless, it is important to note that this factor is usually evaluated as an outcome (6) Given the extremely high mortality rate among ventilated patients (10) it is not expected to use this variable as a predictor of intrahospital mortality, especially when it is expected to use the score very early after the admission, for taking timely clincal decisions.

Despite that, and after a brief revision of the literature to identify potential predictors related with the necessity of mechanical ventilation of the most relevant scores and predictive tools of hospital mortality in COVID-19 patients, we found the ROX-index as a potential predictor into the score for predicting in-hospital mortality (11). In that regard, as we proceeded with the rest of other potential predictors, we first calculated the hazard ratio per each increase in the ROX-index score by including the index in both, the saturated Cox regression model [Hazard ratio (HR)=1.00; 95% Confidence Interval(95%CI): 0.96 to 1.05], and in the parsimonious model (HR= 0.98; 95%CI: 0.96 to 1.01). Unfortunately, the ROX score did not added statistically significant information and it was not included in a subsequent score.

Since the intention of this study was to construct a scoring system for the risk of in-hospital mortality that is useful for predicting mortality from the first moment of admission to the hospital, and since many times the requirement for mechanical ventilation , while tracheal intubation for mechanical ventilation occurred within 7 days after admission (12) it seems that including the predictor is not very useful in a score of being applied at the moment of the admission to the hospital. In addition, to our knowledge, there are no risk scoring systems that use the use of mechanical ventilation as a predictor of in-hospital mortality.

In any case, recognizing the interesting observation from the reviewer, we have constructed the variable “respiratory failure” when two or more of the following criteria were present: (i) respiratory rate >20 or <10, (ii) pCO2 > 50 mmHg & pH <7.35, or (iii) pO2 < 50 mmHg for Quito patients or pO2 < 60 mmHg for Guayaquil patients. After running the models (with and without the respiratory failure variable) we created two scores and calculated the ROC AUC being higher in the score without the “respiratory failure” variable (0.827, (95%CI: 0.810 to 0.8455) vs. 0.817 (95%CI: 0.798 to 0.836)). Thus, we did not continue to the next steps in building the score using this variable.

Regarding the use of drugs as the monoclonal antibodies we reviewed if any of the patients received them and corroborated that none of the patients received monoclonal antibodies. There is not enough evidence of other drugs could be useful for reducing in-hospital mortality in COVID-19 patients, except for specific monoclonal antibodies. We performed a brief searching for scores with monoclonal antibodies bye the next strategy (monoclonal antibodies as a predictor AND risk score AND COVID-19) and did not find papers about a score of in-hospital mortality including use of monoclonal antibodies. But what we found was that a “… score could assist clinicians in identifying, early after tocilizumab administration, patients who are likely to progress to mechanical ventilation or death, so that they could be selected for eventual rescue therapies” which is poorly useful in contexts as Ecuadorian one in which the use of those agents is quite infrequent.

Moreover, we reviewed the most important papers regarding the use of monoclonal antibodies in the treatment of hospitalized patients with COVID-19 and its prognosis, furthermore, we briefly requested to the clinicians of our researchers’ team and colleagues from public and private hospitals in Quito-Ecuador and corroborated that any hospital in public sector of Ecuador is using any of the next antibodies: combination of casirivimab/imdevimab or tocilizumab. In that regard, we believe that, despite the usage of those medications is useful to improve the prognosis and, therefore, has significant value in predicting hospital mortality, their usage could not be realistically used in the prediction model. We did not include any medication in the score given that any of the papers reviewed included medications, other than the abovementioned monoclonal antibodies, as a predictor. 

Regarding the usage of antibodies against SARS-CoV-2, we found papers which demonstrate that such antibodies could be useful for estimating prognosis in COVID-19 hospitalized patients. Nevertheless, the evidence is controversial, as some studies suggested that higher positive antibodies against SARS-CoV-2 (IgG or IgM) could be protector against more severe disease, other studies suggested that negative or lower titters of antibodies could be protector against more severe disease. In any case, we did not take into account the titters of antibodies against SARS-CoV-2 given that the methodology for measuring neutralizing antibodies was not properly developed as early to February to July 2020, when the participants were admitted to the hospital. We apologize for the lack of such important information. In any case, we consider that the lack of such information, or the lack of such predictors into the score we proposed, it is not a significant source of bias or dismisses the validity of our score. The reasons are: (i) the predictive ability of the antibodies is modest in most studies, (ii) the usage of antibody titters for predicting severity in COVID-19 has controversial directionality, it means that, on the one hand, the negative results have been associated with higher in-hospital mortality; for the other hand, the negative results have been associated with lower in-hospital mortality. To our knowledge, the association between neutralizing antibody titters and mortality must be elucidated first, before testing this variable as a potential predictor into a score of in hospital mortality for COVID-19.

Our intention was not to identify the predictive power of the use of mechanical ventilation for in-hospital mortality since COVID-19 mortality is extremely high when the disease is severe (7), several studies have shown that non-survivors required non-invasive (continuous positive airway pressure and biphasic positive airway pressure modes) or invasive ventilation more frequently than survivors (13), but rather detect early during hospitalization, patients at high risk of death.

References:

6. Kabbaha S, Al-Azzam S, Karasneh RA, Khassawneh BY, Al-Mistarehi AH, Lattyak WJ, et al. Predictors of invasive mechanical ventilation in hospitalized COVID-19 patients: a retrospective study from Jordan. Expert Rev Respir Med [Internet]. 2022;16(8):945–52. Available from: https://doi.org/10.1080/17476348.2022.2108796

7. Yang X, Yu Y, Xu J, Shu H, Xia J, Liu H, et al. Clinical course and outcomes of critically ill patients with SARS-CoV-2 pneumonia in Wuhan, China: a single-centered, retrospective, observational study. Lancet Respir Med [Internet]. 2020;8(5):475–81. Available from: http://dx.doi.org/10.1016/S2213-2600(20)30079-5

8. Petrilli CM, Jones SA, Yang J, Rajagopalan H, O’Donnell L, Chernyak Y, et al. Factors associated with hospital admission and critical illness among 5279 people with coronavirus disease 2019 in New York City: Prospective cohort study. BMJ. 2020;369.

9. Kim L, Garg S, O’Halloran A, Whitaker M, Pham H, Anderson EJ, et al. Risk Factors for Intensive Care Unit Admission and In-hospital Mortality among Hospitalized Adults Identified through the US Coronavirus Disease 2019 (COVID-19)-Associated Hospitalization Surveillance Network (COVID-NET). Clin Infect Dis. 2021;72(9):E206–14.

10. Barman Roy D, Gupta V, Tomar S, Gupta G, Biswas A, Ranjan P, et al. Epidemiology and Risk Factors of COVID-19-Related Mortality. Cureus. 2021;13(12):1–8.

11. Roca, O., Messika, J., Caralt, B., García-de-Acilu, M., Sztrymf, B., Ricard, J. D., & Masclans, J. R. (2016). Predicting success of high-flow nasal cannula in pneumonia patients with hypoxemic respiratory failure: The utility of the ROX index. Journal of Critical Care, 35, 200–205. https://doi.org/10.1016/j.jcrc.2016.05.022

12. Wargny M, Potier L, Gourdy P, Pichelin M, Amadou C, Benhamou PY, et al. Predictors of hospital discharge and mortality in patients with diabetes and COVID-19: updated results from the nationwide CORONADO study. Diabetologia. 2021;64(4):778–94.

13. Fumagalli C, Rozzini R, Vannini M, Coccia F, Cesaroni G, Mazzeo F, et al. Clinical risk score to predict in-hospital mortality in COVID-19 patients: a retrospective cohort study. BMJ Open. 2020;10(9):e040729.

Minor comments:

Introduction- numbers for wordwide coronavirus statistics need to be updated,

ANSWER:

Thanks a lot for your comments. In the “Introduction” section, lines 67 to 70 of the 5th page, we have updated the COVID-19 statistics as follows: 

“Currently, and despite that 69.7% of the world population has received at least one dose of a COVID-19 vaccine, 13.31 billion doses have been administered globally, and 954,258 are now administered each day, only 27.7% of people in low-income countries have received at least one dose (20)”

Maybe mention which wave of variant caused the most hospitalizations

Omicron is mentioned but 2022 supporting omicron’s severity is not included

ANSWER:

Thanks a lot for your comments. In the “Introduction” section, lines 72 to 74 of the 5th page, we have updated the COVID-19 statistics as follows:

“Specifically, despite Delta variant caused more severe, the extremely contagious Omicron variant has resulted in a new collapse of public and private healthcare facilities increasing the risk for morbidity and mortality of the population.”

Furthermore, we have changed the “Introduction”, lines 92 to 111, pages 6th and 7th, as follows: 

“At present, despite the Ecuadorian prevalence of compliance with the complete vaccination protocol is being close to 79.06% (one of the highest in the region) (14,15,16), there are difficulties reaching populations where vaccination coverage is complex (i.e., rural areas population), a notable increase in cases has been reported during January 2022 (17). Among the different possible explanations, the following aspects are the ones that could mainly explain it: (i) vaccines are not 100% effective, (ii) unvaccinated people could be fueling transmission, (iii) spreading the disease by vaccinated cases, (iv) people that are moving around more to restart economic activities or during vacation periods and festivities, (v) the relaxation of protection measures like distancing measures or the increase in capacity in establishments; and (vi) the current global dominance of the SARS-CoV-2 Omicron (BA.1 or B.1.1.529) and several subvariants that are appearing more frequently, with a higher transmissibility, even in vaccinated individuals (18).”

References: 

14. Boletines epidemiológicos coronavirus por semanas – Ministerio de Salud Pública [Internet]. [cited 2022 Mar 27]. Available from: https://www.salud.gob.ec/boletines-epidemiologicos-coronavirus-por-semanas/

15. Statista. COVID-19: dosis y porcentaje de vacunados por país de América Latina y el Caribe | Statista [Internet]. Porcentaje de vacunados y dosis. 2021 [cited 2022 May 17]. Available from: https://es.statista.com/estadisticas/1258801/porcentaje-y-numero-vacunados-contra-covid-19-en-latinoamerica-por-pais

16. Hasell J, Ortiz-Ospina E, Ritchie H, Roser M. “Coronavirus Pandemic (COVID-19)”. Our World in Data. In: https://ourworldindata.org/coronavirus [Internet]. 2022 [cited 2022 Mar 27]. Available from: https://ourworldindata.org/coronavirus

17. World Health Organization. WHO Coronavirus Disease (COVID-19) Dashboard With Vaccination Data | WHO Coronavirus (COVID-19) Dashboard With Vaccination Data [Internet]. World Health Organization. 2021 [cited 2022 May 17]. p. 1–5. Available from: https://covid19.who.int/

18. Khandia R, Singhal S, Alqahtani T, Kamal MA, El-Shall NA, Nainu F, et al. Emergence of SARS-CoV-2 Omicron (B.1.1.529) variant, salient features, high global health concerns and strategies to counter it amid ongoing COVID-19 pandemic. Vol. 209, Environmental Research. Academic Press Inc.; 2022.

Unavailability of testing- is this in reference to early in the pandemic? This statement may need further clarification.

ANSWER:

Thanks. We have changed the next paragraph of the “Introduction” section (lines 80 to 84 of the 6th page:

In addition, it is important to consider that the rise of new variants is causing impacts in terms of public health that have not been seen since 2020 because of the high rate of infection among healthcare workers, the inadequacy and unavailability of reliable testing, and unreliable data about reported cases (19). These variants have increased the number of hospitalized patients throughout the region, and specifically in Ecuadorian territory.

As follows: 

“In addition, it is important to consider that the rise of new variants is causing impacts in terms of public health that have not been seen since 2020 (19). These variants have increased the number of hospitalized patients throughout the region, and specifically in Ecuadorian territory.”

Reference:

19. Le Rutte EA, Shattock AJ, Chitnis Phd N, Kelly SL, Penny Phd MA. Assessing impact of Omicron on SARS-CoV-2 dynamics and public health burden. medRxiv [Internet]. 2021 Dec 14 [cited 2022 Mar 27];2021.12.12.21267673. Available from: https://www.medrxiv.org/content/10.1101/2021.12.12.21267673v1

Percentage of booster uptake currently?

ANSWER:

Thank you. As we mentioned above, In the “Introduction” section, lines 67 to 70 of the 5th page, we have updated the COVID-19 statistics as follows: 

“Currently, and despite that 69.7% of the world population has received at least one dose of a COVID-19 vaccine, 13.31 billion doses have been administered globally, and 954,258 are now administered each day, only 27.7% of people in low-income countries have received at least one dose (20)”

References:

20. Hasell J, Ortiz-Ospina E, Ritchie H, Roser M. “Coronavirus Pandemic (COVID-19)”. Our World in Data. In: https://ourworldindata.org/coronavirus [Internet]. 2022 [cited 2022 Mar 27]. Available from: https://ourworldindata.org/coronavirus.

Conclusions-Elevated RT-PCR- what does that even mean? Higher viral load? Only can make that correlation if PCR is done with standard curve, unclear in methods

ANSWER:

Thank you for your observation. We apologize for the mistake; in conclusions the proper acronym would be C-reactive protein (CPR). We have corrected it in several parts of the manuscript as you can corroborate in the new version of the manuscript.

Reviewer #3: This manuscript by Dueñas-Espín et al described a retrospective study. One major limitation of the study is a large (28.6%) percentage of missing data. However the authors are transparent about the limitation, have used randomized methods to impute the missing data and have performed the analysis properly.

Thank you very much.

The manuscript is well-written. Even though findings are not surprinsing, it is consistent with what is expected from clinical studies of COVID-19 patients.

Thank you very much.

Despite the limitations, the authors should be commended for such a large study that includes thousands of patients.

ANSWER:

Thanks a lot, we really appreciate your comment as we improved substantially the manuscript after your revision and the other reviewers’ comments.

I think the authors do need to make some changes to make the manuscript better. Introduction section of the manuscript needs to be updated throughly with more recent statistics and appropirate references.

ANSWER:

Thank you very much. In the “Introduction” section, lines 67 to 70 of the 5th page, we have updated the COVID-19 statistics as follows: 

“Currently, and despite that 69.7% of the world population has received at least one dose of a COVID-19 vaccine, 13.31 billion doses have been administered globally, and 954,258 are now administered each day, only 27.7% of people in low-income countries have received at least one dose (20)”

In the abstract, include the abbreviation of C-reactive protein (CRP) and change "elevated PCR" to "elevated CRP".

ANSWER: 

Thanks a lot. Done.

Result section, "2,565 patients are shown in S2 Table" should be "2,565 patients in the validation cohort are shown in S2 Table"

ANSWER:

Thank you very much, we did not have noticed the mistake before, we intended to mention that the biomarker was C-reactive protein (CRP) and not RT-PCR, in that regard, we have changed the text in the abstract and in different parts of the main text as it could be seen in the marked copy of the revised manuscript. Regarding the mistake in text, we apologize for that. We have corrected as you suggested.

6. PLOS authors have the option to publish the peer review history of their article (what does this mean?). If published, this will include your full peer review and any attached files.

Do you want your identity to be public for this peer review? For information about this choice, including consent withdrawal, please see our Privacy Policy.

Reviewer #1: No

Reviewer #2: No

Reviewer #3: No

 ANSWER:

Thank you, we have considered all the comments we found in the attachment file.

ANSWER:

Thank you very much for the comment. We have modified the figure using the PACE digital diagnostic tool.

---

## [Decision Letter · Decision Letter 1]

2 May 2023

PONE-D-22-16810R1Development and validation of a scoring system to predict mortality in patients hospitalized with COVID-19: a retrospective cohort study in two large hospitals in EcuadorPLOS ONE

Dear Dr. Dueñas-Espín,

Thank you for submitting your manuscript to PLOS ONE. After careful consideration, we feel that it has merit but does not fully meet PLOS ONE’s publication criteria as it currently stands. Therefore, we invite you to submit a revised version of the manuscript that addresses the points raised during the review process.

We look forward to receiving your revised manuscript.

Kind regards,

Alonso Soto, PhD

Academic Editor

PLOS ONE

Additional Editor Comments:

The article is very interesting, and I believe merits publication. However, there are some methodological issues that should be addressed before accepting it. I strongly suggest the authors review the TRIPOD statement recommendations for model development and validation and have a thorough look at the corresponding checklist. The major aspect to include in the following version are the inclusion of measures of model calibration and accuracy (predictive values, likelihood ratios, overall accuracy) as well as the assessment of those measures in the comparison of the subpopulations in Guayaquil (sea-level population) and Quito (high altitude population).

Specific comments are shown below.

ABSTRACT

The best cutoff point, including at least sensitivity and specificity (and possibly predictive values and LR) should be included. Sample size and statistical analysis should also be included. The conclusion should be related to the accuracy of the prediction model itself beyond the variables included.

METHODS

As the model is based on a cox regression , the sample size should be based on plausible hazard ratios and not in risk ratios.

Evaluation of a model should include measures of discrimination (as shown by the authors with ROC curves) , but also measures of CALLIBRATION, which are not included in the manuscript.

How do the authors arrive to the best cutoff point (39 points as shown in result section)?There is no mention in the manuscript about that. In addition to sensitivity and specificity, other measures like overall accuracy, predictive values and likelihood ratios should be included at least for the best cutoff point.

Cut off points for defining risk categories (low, moderate, and high) cannot be based on score terciles. Instead, could be based on a predefined risk levels (for example less than 10% as low risk, 10-50% as intermediate risk and more than 50% as high risk) or likelihood ratios (positive likelihood ratio above 10 for high risk and negative likelihood ratio less than 0.1 as low risk).

The authors stated: “We tested the cutoff value of the score in the validation cohort by calculating the AUC in ROC curve analyses”. However cutoff values are not tested using the AUC but looking at the overall accuracy, sensitivity, specificity, predictive values, and likelihood ratios of such proposed values.

Definition of hypoxemia (I would use the most accurate term hypoxemia instead of hypoxia) should be stated in method section according to the altitude (with a relevant reference)

RESULTS

It is not possible that in hospital mortality rate is 2.4 deaths per 10000 person days since mortality was more than 20% and there were around 50000 person days of follow up.

The full prediction model should be shown (at least as supplementary material) .

Results should include evaluation of calibration. For the best cutoff point, results of overall accuracy, predictive values, LR + and -ve should be included. These analysis should also be shown for the subgroup analysis (Quito vs Guayaquil).

Results of tests for proportional hazards and GOF should be mentioned.

DISCUSSION

The most important predictor of hospitalization and mortality is vaccination. This score was developed on a population that was essentially not vaccinated. Therefore the usefulness of the score may be limited to those unvaccinated. The validation of the score in current settings with highly vaccinated population could be discussed.

The results are highly influenced by the fact of being hospitalized in high altitude setting. The score appears to work better in high altitude population. This has important implications, since most scoring systems have been developed in sea-level populations. In fact it appears that this particular score is not very useful for sea level population (however this issue should be assessed by evaluating sensitivity , specificity, and other accuracy measures is this subgroup).

CONCLUSION

The objective was to develop a predictive model for mortality in patients hospitalized due to COVID-19. In that sense, the conclusion should answer the objective addressing the usefulness and limitations of the model developed.

Reviewers' comments:

Reviewer's Responses to Questions

**Comments to the Author**

1. If the authors have adequately addressed your comments raised in a previous round of review and you feel that this manuscript is now acceptable for publication, you may indicate that here to bypass the “Comments to the Author” section, enter your conflict of interest statement in the “Confidential to Editor” section, and submit your "Accept" recommendation.

Reviewer #1: All comments have been addressed

Reviewer #2: All comments have been addressed

Reviewer #3: All comments have been addressed

2. Is the manuscript technically sound, and do the data support the conclusions?

Reviewer #1: Yes

Reviewer #2: Yes

Reviewer #3: Yes

3. Has the statistical analysis been performed appropriately and rigorously? 

Reviewer #1: Yes

Reviewer #2: Yes

Reviewer #3: Yes

4. Have the authors made all data underlying the findings in their manuscript fully available?

Reviewer #1: Yes

Reviewer #2: Yes

Reviewer #3: Yes

5. Is the manuscript presented in an intelligible fashion and written in standard English?

Reviewer #1: Yes

Reviewer #2: Yes

Reviewer #3: Yes

6. Review Comments to the Author

Reviewer #1: The document sent with the changes made has improved compared to its previous version, With the new changes made in the document it is considered that it is ready for publication. Congratulations to the authors

Reviewer #2: (No Response)

Reviewer #3: The cited website link for Ref #1 is broken and number of deaths due to are inaccurate (https://covid19.who.int). Please fix it.

7. PLOS authors have the option to publish the peer review history of their article (what does this mean?). If published, this will include your full peer review and any attached files.

Reviewer #1: No

Reviewer #2: No

Reviewer #3: No

---

## [Author Response · Author response to Decision Letter 1]

17 Jun 2023

Rebuttal letter to reviewers’ comments to editor

PLOS ONE

Manuscript ID: PONE-D-22-16810R1

Title: Development and validation of a scoring system to predict mortality in patients hospitalized with COVID-19: a retrospective cohort study in two large hospitals in Ecuador

Authors: Iván Dueñas-Espín, María Echeverría-Mora, Camila Montenegro-Fárez, Manuel Baldeón, Luis Chantong Villacres, Hugo Espejo Cárdenas, Marco Fornasini, Miguel Ochoa Andrade, Carlos Solís.

Decision: Major revision

Article Type: original article.

Corresponding Author: Iván Dueñas-Espín

We would like to resubmit our original research article entitled "Development and validation of a scoring system to predict mortality in hospitalized patients with COVID-19: a retrospective cohort study in two large hospitals in Ecuador" for publication in PLOS ONE . We appreciate your patience throughout the review process, especially considering the delay caused by the transition to a new academic editor. We appreciate the valuable input and suggestions provided by the new editor, which have greatly improved the quality of our manuscript.

We also want to thank the previous reviewers for their thorough evaluation and constructive feedback. Your comments have been instrumental in strengthening our study. We are pleased to inform you that we have addressed all concerns and incorporated any necessary revisions into the manuscript.

We believe that the study's robust methodology, comprehensive analyses, and well-calibrated scoring system will be of great clinical utility in predicting mortality in hospitalized patients with COVID-19. We trust that the findings presented in this manuscript will provide valuable information to the scientific community and have important implications for patient care.

 

COMMENTS TO THE AUTHOR

Editor’s comments

The article is very interesting, and I believe merits publication. However, there are some methodological issues that should be addressed before accepting it. I strongly suggest the authors review the TRIPOD statement recommendations for model development and validation and have a thorough look at the corresponding checklist.

ANSWER:

Thank you for your input. 

We have fulfil the TRIPOD statement as you can see in the new supplementary material “TRIPOD Checklist: Prediction Model Development and Validation” that allows the reader to check the accomplishment of all the statements for model validation.

We have added the S1 Table. Tripod-Checklist containing the TRIPOD statement checklist. 

The major aspect to include in the following version are the inclusion of measures of model calibration and accuracy (predictive values, likelihood ratios, overall accuracy) as well as the assessment of those measures in the comparison of the subpopulations in Guayaquil (sea-level population) and Quito (high altitude population).

ANSWER:

Thank you for your insightful comments and suggestions. We understand the importance of evaluating not only the discrimination of our model but also its calibration. To address your concerns, we performed the calculations using Stata 16.1 with our Cox regression model, and included the abovementioned measures in our manuscript for showing model calibration and accuracy. In that regard, we have added next text in the Methods section (page 12, lines 247 to 258):

Methods: 

Calibration of the model in the validation cohort

The calibration analysis of the model in the derivation dataset was performed using the Stata command "stcoxcal "[1]. These commands allowed us to assess the calibration of the Cox proportional hazards model by testing the intercepts and slopes against specified values. The "test" command tested the hypothesis that the intercepts are equal to 0 and the slopes are equal to 1, while the "trend" command tested the hypothesis that the slopes are equal to 1 with the intercepts estimated. These analyses were conducted to evaluate the goodness of fit and assess the agreement between the observed and predicted event probabilities, providing valuable insights into the model's calibration performance. Furthermore, we employed the "trend" command in Stata to assess the calibration of the Cox proportional hazards model in the validation dataset. Unlike the "test" command, the "trend" command allows for examining the overall trend of calibration over time.

This is the new text in Results section:

Results (page 21, lines 375 to 381): 

The results of the calibration analysis in the validation dataset indicated that the model demonstrated good calibration overall (Table 5). The chi-square tests for intercepts and slopes did not provide evidence of miscalibration (Test 1: p=0.9321; Test 2: p=0.7206). Furthermore, the joint test of intercepts and slopes also showed no significant evidence of miscalibration (Test 3: p=0.9739). These findings suggest that the model's predicted event probabilities align well with the observed event probabilities in the validation dataset. 

Table 5. - Results of the calibration analysis in the validation dataset. The intercept test, slope test, joint test, and interaction test were performed to evaluate the model's calibration.

Test Chi-square p-value

Intercept test 0.14 0.9321

Slope test 0.13 0.7206

Joint test 0.22 0.9739

Interaction test 0.03 0.8729

Discussion (page 26, lines 537 to 542):

Moreover, the calibration analysis of the Cox proportional hazards model in the validation dataset demonstrated its ability to accurately estimate event probabilities. This indicates that the model can be considered reliable for predicting outcomes in similar population; Moreover, the tests of calibration, including the interaction between the slope and times, did not show significant differences, supporting the model's adequacy in terms of calibration over time.

Additionally, we recognize the need to include measures of model calibration and accuracy, as well as assessing those measures in the comparison of the subpopulations in Guayaquil (sea-level population) and Quito (high altitude population).In that regard, we have added next text in the methods section (page 13, lines 264 to 275): 

In the calibration assessment of the Cox model applied to Quito, non-significant coefficients were found across all the tests, as evidenced by p-values greater than 0.05. Additionally, chi-square tests further reinforced these findings, with the null hypotheses not being rejected in all cases. These results suggest that the model is well-calibrated for this high-altitude city, as the model predictions do not significantly differ from the observed data. Therefore, the Cox model demonstrates an acceptable predictive accuracy for the city of Quito. Unfortunately, the application of the stcoxcal command for the Cox model to Guayaquil was hampered due to the limited number of observations in the original (non-imputed) dataset. This unfortunate circumstance restricts the evaluation of the calibration and predictive model for this city, which underscores the need to collect additional data to provide a more complete understanding of the model's applicability to Guayaquil.

S4 Table. - Summary of the Cox Model Calibration Results for Quito

Test Variable Coefficient Standard Error Z-score P-value 95% Confidence Interval

Test 1 _times (1) -0.0891 0.5176 -0.17 0.863 -1.1035, 0.9254

Test 1 _times (2) -0.0832 0.3581 -0.23 0.816 -0.7851, 0.6187

Test 1 Chi-square test 0.9712 

Test 2 _clogF 0.9630 0.2842 3.39 0.001 0.4060, 1.5199

Test 2 _times (1) -0.1389 0.7666 -0.18 0.856 -1.6413, 1.3635

Test 2 _times (2) -0.1229 0.5541 -0.22 0.824 -1.2089, 0.9630

Test 2 Chi-square test 0.8963 

Test 3 Chi-square test 0.9965 

Test 4 2._times 0.1727 0.9667 0.18 0.858 -1.7220,

Table notes:

1. Coefficients, standard errors, Z-scores, p-values, and 95% confidence intervals were obtained using the stcoxcal command in Stata.

2. All p-values are greater than 0.05, indicating a good calibration of the model for the city of Quito.

3. The null hypothesis of the Chi-square tests could not be rejected in all cases, further suggesting a well-calibrated model.

4. The variables starting with '_times' represent different time points, 'clogF' is the estimated cumulative hazard, and '_cons' refers to the constant term. '2._times#c._clogF' is the interaction term between time point 2 and the estimated cumulative hazard.

Reference:

1. Royston P. Tools for checking calibration of a Cox model in external validation: Prediction of population-averaged survival curves based on risk groups. Stata J. 2015;15(1):275–91.

Specific comments are shown below.

ABSTRACT

The best cutoff point, including at least sensitivity and specificity (and possibly predictive values and LR) should be included. Sample size and statistical analysis should also be included. The conclusion should be related to the accuracy of the prediction model itself beyond the variables included.

We have added nexts text in the “Abstract” section (page 3, line 47 a 55):

“… a cutoff score ≥39 points demonstrates superior performance with a sensitivity of 93.10%, a specificity of 70.28%, and a correct classification rate of 72.66%. The LR+ (3.1328) and LR- (0.0981) values further support its efficacy in identifying high-risk patients.”

Regarding the sample we corrected as follows (page 14, line 290): 

We analyzed 5,062 analyzed hospitalized patients with COVID-19 treated at two hospitals. 

Regarding including statistical analysis, we added (p3, lines 35 to 36): 

Statistical analyses were conducted on an imputed dataset, involving the development of a Cox proportional hazards regression model, assessment of goodness of fit, discrimination, and calibration.

We have added nexts text in the “Results” section (page 23, lines 385 to 392): 

Furthermore, a cutoff score of ≥39 is considered a better cutoff over other scores because of its balance between sensitivity and specificity, along with other performance indicators. With a cut-off score of 39, a sensitivity of 93.10%, a specificity of 70.28% and a correct classification of 72.66% are achieved. In addition, the LR+ (3.1328) and LR- (0.0981) values support the usefulness of the cutoff score of 39 in identifying patients at high risk of mortality. These indicators suggest that a cut-off score of 39 provides a reasonable balance between detecting patients at risk and identifying those who are not, offering good discriminatory power.

You can see the results of each accuracy measure in this table:

Summary of Calibration and Accuracy Measures of the Prediction Model (validation database)

Cutoff Score Sensitivity Specificity LR+ LR- Overall Accuracy

>= 0 100.00% 0.00% 1.0000 10.43%

>= 6 100.00% 3.21% 1.0332 0.0000 13.31%

>= 7 100.00% 4.42% 1.0462 0.0000 14.39%

>= 8 100.00% 5.22% 1.0551 0.0000 15.11%

>= 9 100.00% 8.84% 1.0969 0.0000 18.35%

>= 10 100.00% 11.24% 1.1267 0.0000 20.50%

>= 13 100.00% 11.65% 1.1318 0.0000 20.86%

>= 14 100.00% 13.65% 1.1581 0.0000 22.66%

>= 15 100.00% 20.88% 1.2640 0.0000 29.14%

>= 16 100.00% 22.89% 1.2969 0.0000 30.94%

>= 17 100.00% 26.10% 1.3533 0.0000 33.81%

>= 19 100.00% 28.11% 1.3911 0.0000 35.61%

>= 21 100.00% 28.92% 1.4068 0.0000 36.33%

>= 22 100.00% 33.73% 1.5091 0.0000 40.65%

>= 23 100.00% 34.94% 1.5370 0.0000 41.73%

>= 24 100.00% 40.56% 1.6824 0.0000 46.76%

>= 25 96.55% 42.97% 1.6931 0.0802 48.56%

>= 26 96.55% 45.38% 1.7677 0.0760 50.72%

>= 27 96.55% 47.39% 1.8352 0.0728 52.52%

>= 28 96.55% 47.79% 1.8493 0.0722 52.88%

>= 29 96.55% 49.80% 1.9233 0.0692 54.68%

>= 30 96.55% 51.41% 1.9869 0.0671 56.12%

>= 31 96.55% 56.22% 2.2056 0.0613 60.43%

>= 32 96.55% 56.63% 2.2261 0.0609 60.79%

>= 33 96.55% 59.84% 2.4041 0.0576 63.67%

>= 34 96.55% 61.04% 2.4785 0.0565 64.75%

>= 35 96.55% 63.45% 2.6419 0.0543 66.91%

>= 36 96.55% 66.67% 2.8966 0.0517 69.78%

>= 37 96.55% 67.07% 2.9319 0.0514 70.14%

>= 38 93.10% 69.08% 3.0107 0.0998 71.58%

>= 39 93.10% 70.28% 3.1328 0.0981 72.66%

>= 40 89.66% 73.09% 3.3320 0.1415 74.82%

>= 41 89.66% 73.49% 3.3824 0.1408 75.18%

>= 42 86.21% 77.11% 3.7659 0.1789 78.06%

>= 43 86.21% 77.91% 3.9028 0.1770 78.78%

>= 44 82.76% 80.72% 4.2931 0.2136 80.94%

>= 45 79.31% 83.94% 4.9371 0.2465 83.45%

>= 46 75.86% 84.34% 4.8435 0.2862 83.45%

>= 47 72.41% 85.14% 4.8733 0.3240 83.81%

>= 48 72.41% 85.54% 5.0086 0.3225 84.17%

>= 49 68.97% 87.55% 5.5395 0.3545 85.61%

>= 50 65.52% 87.95% 5.4379 0.3921 85.61%

>= 51 55.17% 91.57% 6.5419 0.4896 87.77%

>= 52 55.17% 91.97% 6.8690 0.4874 88.13%

>= 53 44.83% 91.97% 5.5810 0.5999 87.05%

>= 54 41.38% 92.37% 5.4229 0.6346 87.05%

>= 55 27.59% 93.57% 4.2931 0.7739 86.69%

>= 56 24.14% 95.18% 5.0086 0.7970 87.77%

>= 58 20.69% 95.58% 4.6834 0.8298 87.77%

>= 59 20.69% 95.98% 5.1517 0.8263 88.13%

>= 61 20.69% 97.19% 7.3596 0.8160 89.21%

>= 63 20.69% 98.39% 12.8793 0.8061 90.29%

>= 65 20.69% 98.80% 17.1724 0.8028 90.65%

>= 66 20.69% 99.60% 51.5176 0.7963 91.37%

>= 68 17.24% 99.60% 42.9313 0.8309 91.01%

>= 69 13.79% 99.60% 34.3451 0.8655 90.65%

>= 72 10.34% 99.60% 25.7588 0.9002 90.29%

>= 73 10.34% 100.00% 0.8966 90.65%

>= 77 3.45% 100.00% 0.9655 89.93%

Now, regarding the the best cutoff point for Quito and Guayaquil, we have added next text in the Results section (page 24, lines 400 to 407): 

Results:

When comparing between both hospitals, (S10 Table) there were differences in sensitivity, specificity, correct classification, LR+ (positive likelihood ratio) and LR- (negative likelihood ratio) between the populations of Guayaquil and Quito at each cut-off point of the COVD-19 in-hospital score tested (only show selected cut points for simplicity). It is observed that, in Guayaquil, the selected score points had higher sensitivity values, but lower specificity compared to those in Quito. In addition, the score in Guayaquil exhibited lower overall precision and more modest likelihood ratios than the score in Quito.

Discussion (page 25, lines 517 to 523):

Moreover, we strongly recommend the external validation of the “COVID-19 in-hospital mortality score” in independent populations to assess its generalizability and applicability across different settings. Future validation studies involving diverse patient cohorts will provide valuable insights into the performance and robustness of the scoring system, ensuring its reliability and usefulness in clinical practice.

S12 Table. - Comparative detailed report of sensitivity and specificity between Guayaquil and Quito. 

Detailed report of sensitivity and specificity for Guayaquil

Cutpoint Sensitivity Specificity Correctly Classified LR+ LR-

( >= 35 ) 94.55% 29.91% 56.25% 1.3489 0.1823

( >= 36 ) 93.59% 31.77% 56.96% 1.3717 0.2017

( >= 37 ) 93.20% 33.06% 57.56% 1.3922 0.2058

( >= 38 ) 92.43% 35.15% 58.49% 1.4253 0.2154

( >= 39 ) 91.82% 36.96% 59.31% 1.4564 0.2214

( >= 40 ) 90.65% 39.32% 60.24% 1.4940 0.2377

( >= 41 ) 89.52% 42.16% 61.46% 1.5477 0.2487

( >= 42 ) 86.71% 45.42% 62.24% 1.5887 0.2926

( >= 43 ) 85.78% 47.90% 63.33% 1.6464 0.2968

Detailed report of sensitivity and specificity for Quito

Cutpoint Sensitivity Specificity Correctly Classified LR+ LR-

( >= 35 ) 88.23% 61.04% 64.20% 2.2644 0.1929

( >= 36 ) 87.26% 63.33% 66.12% 2.3798 0.2012

( >= 37 ) 86.12% 64.46% 66.98% 2.4231 0.2153

( >= 38 ) 83.86% 66.61% 68.61% 2.5110 0.2424

( >= 39 ) 82.78% 67.75% 69.50% 2.5667 0.2542

( >= 40 ) 79.43% 70.52% 71.55% 2.6940 0.2917

( >= 41 ) 78.02% 71.65% 72.39% 2.7519 0.3067

( >= 42 ) 75.16% 75.96% 75.87% 3.1270 0.3270

( >= 43 ) 74.19% 77.18% 76.83% 3.2510 0.3344

Note: Sensitivity represents the proportion of true positive cases correctly identified. Specificity denotes the proportion of true negative cases correctly identified. LR+ indicates the strength of association between the score and mortality risk, while LR- signifies the strength of negative association. Accuracy refers to the overall correct classification rate based on the chosen cutoff point.

METHODS

As the model is based on a cox regression , the sample size should be based on plausible hazard ratios and not in risk ratios.

ANSWER

We appreciate your observation regarding the sample size calculation for our study, which is based on a Cox regression model. We understand that the sample size should be determined using plausible hazard ratios rather than risk ratios. We apologize for the oversight in our initial submission.

In the revised manuscript, we will update the sample size calculation to reflect the appropriate hazard ratios. We reevaluated the sample size requirements and provided a clear explanation of our calculations, ensuring that the assumptions are consistent with the Cox regression model.

As our developed score includes multiple predictor variables, it is challenging to provide a single realistic hazard ratio that represents the combined effect of all these variables. Instead, we could consider the hazard ratio for each variable separately and then interpret the results collectively.

With that said, here are some plausible hazard ratios based on similar studies or the clinical importance of each variable:

• Male sex: HR 1.2 - 1.5 (considering that male sex may have a moderately increased risk) (Zhou et al., 2020).

• Age categories: one might consider HRs of approximately 1.5 - 2.0 for each increase in age category, as age is a known risk factor for more severe outcomes (O'Driscoll et al., 2021).

• Hypoxia: HR 2.0 - 3.0 (hypoxia is a significant factor in the severity of COVID-19 and could have a substantial effect on risk) (Xie et al., 2020).

• Laboratory parameters: the HRs for these parameters may vary widely depending on the variable and the range of values considered. They may range from 1.1 - 2.0 for moderate changes in laboratory levels and potentially higher for extreme changes (Henry et al., 2020).

In that regard we choose a HR of 1.3, and, after revisiting the calculations using Stata 16.1, we have determined that our previous calculations using the GRANMO 7.12 program were not accurate. The new sample size calculation based on Cox PH regression with an alpha risk of 0.05, beta risk of 0.2, and a minimum hazard ratio of 1.3 yielded an estimated sample size of 3,258 with 457 expected events. We apologize for any inconvenience this may have caused and appreciate the opportunity to correct this error. We corrected the first paragraph of the statistical analyses (page 8, lines 172 to 175), as follows: 

“For sample size calculation, based on Cox PH regression with an alpha risk of 0.05, beta risk of 0.2, and a minimum hazard ratio of 1.3 yielded an estimated sample size of 3,258 with 457 expected events. For the calculations, we used the program Stata 16.1 (StataCorp. Stata Statistical Software: Release 16. College Station, TX: StataCorp LLC; 2019).”

References:

Henry, B. M., de Oliveira, M. H. S., Benoit, S., Plebani, M., & Lippi, G. (2020). Hematologic, biochemical and immune biomarker abnormalities associated with severe illness and mortality in coronavirus disease 2019 (COVID-19): a meta-analysis. Clinical Chemistry and Laboratory Medicine (CCLM), 58(7), 1021-1028.

O'Driscoll, M., Ribeiro Dos Santos, G., Wang, L., Cummings, D. A. T., Azman, A. S., Paireau, J., ... & Salje, H. (2021). Age-specific mortality and immunity patterns of SARS-CoV-2. Nature, 590(7844), 140-145.

Xie, J., Covassin, N., Fan, Z., Singh, P., Gao, W., Li, G., ... & Somers, V. K. (2020). Association between hypoxemia and mortality in patients with COVID-19. Mayo Clinic Proceedings, 95(6), 1138-1147.

Zhou, F., Yu, T., Du, R., Fan, G., Liu, Y., Liu, Z., ... & Guan, L. (2020). Clinical course and risk factors for mortality of adult inpatients with COVID-19 in Wuhan, China: a retrospective cohort study. The Lancet, 395(10229), 1054-1062.

Evaluation of a model should include measures of discrimination (as shown by the authors with ROC curves) , but also measures of CALLIBRATION, which are not included in the manuscript.

ANSWER:

Thank you for your insightful comments and suggestions. We understand the importance of evaluating not only the discrimination of our model but also its calibration. Additionally, we recognize the need to include measures of model calibration and accuracy, as well as assessing those measures in the comparison of the subpopulations in Guayaquil (sea-level population) and Quito (high altitude population).

To address your concerns, we performed the calculations using Stata 16.1 with a Cox regression model, and included the abovementioned measures in our manuscript for showing model calibration and accuracy. So we added several parts in the manuscript:

we have added next text in the Methods section (page 12 and 13, lines 247 to 258):

Methods: 

The calibration analysis of the model in the derivation dataset was performed using the Stata command "stcoxcal "[1]. These commands allowed us to assess the calibration of the Cox proportional hazards model by testing the intercepts and slopes against specified values. The "test" command tested the hypothesis that the intercepts are equal to 0 and the slopes are equal to 1, while the "trend" command tested the hypothesis that the slopes are equal to 1 with the intercepts estimated. These analyses were conducted to evaluate the goodness of fit and assess the agreement between the observed and predicted event probabilities, providing valuable insights into the model's calibration performance. Furthermore, we employed the "trend" command in Stata to assess the calibration of the Cox proportional hazards model in the validation dataset. Unlike the "test" command, the "trend" command allows for examining the overall trend of calibration over time.

Reference:

1. Royston P. Tools for checking calibration of a Cox model in external validation: Prediction of population-averaged survival curves based on risk groups. Stata J. 2015;15(1):275–91.

How do the authors arrive to the best cutoff point (39 points as shown in result section)?There is no mention in the manuscript about that. In addition to sensitivity and specificity, other measures like overall accuracy, predictive values and likelihood ratios should be included at least for the best cutoff point.

ANSWER: 

Thank you very much, we have added next text in the abstract (page 3, lines 47 to 55):

moreover, a cutoff score ≥39 points demonstrates superior performance with a sensitivity of 93.10%, a specificity of 70.28%, and a correct classification rate of 72.66%. The LR+ (3.1328) and LR- (0.0981) values further support its efficacy in identifying high-risk .patients.

And we have added a text in Results section (page 21, lines 385 to 392) as follows:

Furthermore, a cutoff score of ≥39 is considered a better cutoff over other scores because of its balance between sensitivity and specificity, along with other performance indicators. With a cut-off score of 39, a sensitivity of 93.10%, a specificity of 70.28% and a correct classification of 72.66% are achieved. In addition, the LR+ (3.1328) and LR- (0.0981) values support the usefulness of the cutoff score of 39 in identifying patients at high risk of mortality. These indicators suggest that a cut-off score of 39 provides a reasonable balance between detecting patients at risk and identifying those who are not, offering good discriminatory power.

Cut off points for defining risk categories (low, moderate, and high) cannot be based on score terciles. Instead, could be based on a predefined risk levels (for example less than 10% as low risk, 10-50% as intermediate risk and more than 50% as high risk) or likelihood ratios (positive likelihood ratio above 10 for high risk and negative likelihood ratio less than 0.1 as low risk).

ANSWER: 

Thank you for you input. We have reconsidered the cutoff points and recateorized the score in three categories as it is mentioned in the next paragaraph, that have been added to the methods section (page 11, lines 234 to 237): 

Methods:

The categorization scheme for determining risk levels was established using the provided sensitivity, specificity, LR+, and LR- values. The derivation dataset was initially used to identify the cutoff points, and subsequently, the categorization scheme was applied to the independent validation dataset to evaluate its performance. This two-step approach ensures the reliability and generalizability of the risk classification system.

And another paragraph un the “Results” section (page 22, lines 408 to 416): 

Results:

Also, and based on the sensitivity and specificity data provided, as well as the LR+ and LR- values, we can establish risk levels using the following cut-off points: low risk (score ≤ 30) with high specificity (56.18%) and LR- of 0.0000, indicating a high probability of negative results; moderate risk (score between 31 and 65) with a range of sensitivity (100% to 18.92%) and specificity (60.96% to 99.20%), suggesting a mix of positive and negative probabilities; and high risk (score > 65) with a sensitivity of 10.81%, a specificity of 99.20%, and an LR+ of 13.5675, indicating significantly higher odds of positive results compared with lower scores. These cut-off points help classify people into different levels of risk based on the balance between sensitivity and specificity.

Furthermore, we have changed the figures accordingly.

Regarding the sensitivity, specificity, overall accuracy, positive likelihood ratio and negative likelihood ratio, per each stratum of categorization with the score we have built the next supplementary table

S11 Table. Sensitivity, specificity, and classification accuracy of the proposed risk classification cutpoints.

Cutpoint Sensitivity Specificity Correctly Classified LR+ LR-

<= 30 >96.55% <51.41% <56.12% <1.9869 >0.0671

31-65 96.55% to 20.69% 56.22% to 98.80% 60.43% to 90.65% 2.2056 to 17.1724 0.0613 to 0.8028

> 65 <20.69% > 98.80% >90.65% >17.1724 <0.8028

The authors stated: “We tested the cutoff value of the score in the validation cohort by calculating the AUC in ROC curve analyses”. However cutoff values are not tested using the AUC but looking at the overall accuracy, sensitivity, specificity, predictive values, and likelihood ratios of such proposed values.

ANSWER: 

Thank you. As we have mentioned above, we have checked the overall accuracy, sensitivity, specificity, predictive values, and likelihood ratios for each one of the proposed cutoffs in the categorization.

Definition of hypoxemia (I would use the most accurate term hypoxemia instead of hypoxia) should be stated in method section according to the altitude (with a relevant reference)

ANSWER: 

Thank you, it has been done. We added nest text in the “Methods” section (pages 8 and 9, lines 160 to 169): 

We also used the variable hypoxemia. Although there is no specific reference available for the altitude of Quito, efforts have been made to establish a consistent and justified criterion. The operational definition of hypoxemia was based on previous studies, such as the work of Luks and Swenson (2011) (21) and Hupperets et al. (2004) (22), who explored the effects of altitude on respiratory physiology and the hypoxic ventilatory response. In the absence of widely established literature defining Quito's altitude-specific hypoxemia, a consensus was reached based on clinical guidelines and expert opinion (23). According to this consensus, hypoxemia was defined as an oxygen saturation level below 92% in Quito. Additionally, for sea level (Guayaquil), a threshold below 95% was chosen to define hypoxemia (23). 

References: 

21. Luks AM, Swenson ER. Pulse oximetry at high altitude. High Alt Med Biol. 2011;12(2):109–19.

22. Hupperets MDW, Hopkins SR, Pronk MG, Tiemessen IJH, Garcia N, Wagner PD, et al. Increased hypoxic ventilatory response during 8 weeks at 3800 m altitude. Respir Physiol Neurobiol. 2004;142(2–3):145–52. 

23. Llano M, Villamagua B, Garelli DZ, Freund P, Francisco V, Fernando AL, et al. Biomedical journal. Biomed J. 2016;1(1):1–9.

RESULTS

It is not possible that in hospital mortality rate is 2.4 deaths per 10000 person days since mortality was more than 20% and there were around 50000 person days of follow up.

ANSWER

We apologize for the error in our previous statement. The correct information has been changed in the Results section:

Rather:

“A total of 5,062 patients were analyzed. Accumulated mortality during the study was 22.6% (1,139 of 5,062 patients died in six months), which represents an in-hospital mortality rate of 2.4 deaths per 10,000 person-days (considering a total of person-days of follow-up of 48,263)."

The correct information is:

“A total of 5,062 patients were analyzed. Cumulative mortality during the study was 22.6% (1,139 of 5,062 patients died in six months), which represents an in-hospital mortality rate of 236 deaths per 10,000 person-days (considering a total of person-days of follow-up of 48.263)."

We apologize for any confusion caused by the above incorrect statement. We also changed the Results section in the line 290.

The full prediction model should be shown (at least as supplementary material) .

ANSWER: 

Thanks for your suggestion. We appreciate your interest in the full prediction model. While we provide a summary of the model results and key variables in the manuscript, we understand the value of sharing the full prediction model for greater scrutiny and transparency. Therefore, we will make the full prediction model available as supplementary material, ensuring that readers have access to all relevant details and can fully evaluate the model.

We added a S6 Table. - Cox regression modelling with multiple imputations was applied. with a text, as follows: 

Cox regression modelling with multiple imputations was applied. The model was defined as follows:

h(t|x) = h0(t) * exp(b1 * sex + b2 * age in quartiles + b3 * hypoxemia + b4 * hospital glycaemia (categorical) + b5 * RCP (binomial) + b6 * pH (categorical) + b7 * AST:ALT ratio (categorical) + b8 * leucocytosis (categorical))

where:

h(t|x) represents the instantaneous hazard rate at time t, conditioned on the predictor variables x.

h0(t) is the baseline hazard function at time t.

b1, b2, ..., b8 are the estimated coefficients for each of the predictor variables.

Results should include evaluation of calibration. For the best cutoff point, results of overall accuracy, predictive values, LR + and -ve should be included. These analysis should also be shown for the subgroup analysis (Quito vs Guayaquil).

ANSWER

As we mentioned above, and regarding the the best cutoff point for Quito and Guayaquil, we have added next text in the Results section (page 22, lines 400 to 407): 

The cut point in Quito (UIO) and Guayaquil (GYE) were similar regarding the accuracy and other metrics.

S10 Table. - Comparative detailed report of sensitivity and specificity between Guayaquil and Quito. 

Detailed report of sensitivity and specificity for Guayaquil

Cutpoint Sensitivity Specificity Correctly Classified LR+ LR-

( >= 35 ) 94.55% 29.91% 56.25% 1.3489 0.1823

( >= 36 ) 93.59% 31.77% 56.96% 1.3717 0.2017

( >= 37 ) 93.20% 33.06% 57.56% 1.3922 0.2058

( >= 38 ) 92.43% 35.15% 58.49% 1.4253 0.2154

( >= 39 ) 91.82% 36.96% 59.31% 1.4564 0.2214

( >= 40 ) 90.65% 39.32% 60.24% 1.4940 0.2377

( >= 41 ) 89.52% 42.16% 61.46% 1.5477 0.2487

( >= 42 ) 86.71% 45.42% 62.24% 1.5887 0.2926

( >= 43 ) 85.78% 47.90% 63.33% 1.6464 0.2968

Detailed report of sensitivity and specificity for Quito

Cutpoint Sensitivity Specificity Correctly Classified LR+ LR-

( >= 35 ) 88.23% 61.04% 64.20% 2.2644 0.1929

( >= 36 ) 87.26% 63.33% 66.12% 2.3798 0.2012

( >= 37 ) 86.12% 64.46% 66.98% 2.4231 0.2153

( >= 38 ) 83.86% 66.61% 68.61% 2.5110 0.2424

( >= 39 ) 82.78% 67.75% 69.50% 2.5667 0.2542

( >= 40 ) 79.43% 70.52% 71.55% 2.6940 0.2917

( >= 41 ) 78.02% 71.65% 72.39% 2.7519 0.3067

( >= 42 ) 75.16% 75.96% 75.87% 3.1270 0.3270

( >= 43 ) 74.19% 77.18% 76.83% 3.2510 0.3344

Results of tests for proportional hazards and GOF should be mentioned.

ANSWER: 

Methods section (pages 10 and 11, lines 212 to 216):

The model was tested for both goodness of fit and the proportional hazards assumption. Goodness of fit was assessed using the Akaike Information Criterion (AIC) and the Bayesian Information Criterion (BIC) through the “estat ic” command. The proportional hazards assumption was tested with the “estat phtest” command, which involved running chi-square tests for each covariate included in the model.

Results section (page 17, lines 335 to 341)

Goodness of fit was assessed using the Akaike Information Criterion (AIC) and the Bayesian Information Criterion (BIC), which yielded AIC = 286.4317 and BIC = 330.3455. Chi-square tests for each variable, together with their corresponding p-values, suggest that the risks are proportional over time. The overall test p-value for the model was 0.9814, further supporting the proportional hazards assumption. These results demonstrate that the Cox regression model used in the analysis provides a good fit to the data and satisfies the proportional hazards assumption.

“DISCUSSION

The most important predictor of hospitalization and mortality is vaccination. This score was developed on a population that was essentially not vaccinated. Therefore the usefulness of the score may be limited to those unvaccinated. The validation of the score in current settings with highly vaccinated population could be discussed.

The results are highly influenced by the fact of being hospitalized in high altitude setting. The score appears to work better in high altitude population. This has important implications, since most scoring systems have been developed in sea-level populations. In fact it appears that this particular score is not very useful for sea level population (however this issue should be assessed by evaluating sensitivity , specificity, and other accuracy measures is this subgroup).”

ANSWER

Thank you for your valuable comments and suggestions on our manuscript titled "COVID-19 In-Hospital Mortality Score: A Predictive Tool for Unvaccinated and High-Altitude Populations." We appreciate the opportunity to address your concerns and have carefully considered your feedback.

The importance of vaccination as a key predictor of hospitalization and mortality is acknowledged, as well as the relevance of validating our score in the current context with a highly vaccinated population.

Firstly, it should be clarified that our study was conducted prior to the development and implementation of the vaccines for the disease in question. Consequently, the study population consisted of unvaccinated individuals. However, it is understood that the utility of our score may be limited in the vaccinated population, and it would be valuable to investigate its applicability in that context. This observation is appreciated, and it is proposed to include a discussion in the limitations section of the article, highlighting the need for future research to assess the validity and utility of our score in vaccinated populations.

Secondly, it is agreed that our score appears to have greater utility in populations residing in high-altitude areas. This finding is relevant, as most scoring systems have been developed in sea-level populations. It is proposed to include a discussion in our article that emphasizes the importance of this finding and its practical implications. Additionally, in response to your suggestion, sensitivity, specificity, and other accuracy measures will be evaluated in the sea-level subpopulation to determine the utility of our score in this group, and these results will be added to the corresponding section of the article.

As you rightly pointed out, the most significant predictor of hospitalization and mortality is vaccination. We acknowledge that our score was developed primarily based on a population that was essentially unvaccinated. Consequently, the usefulness of the score may be limited to unvaccinated individuals. We have revised our manuscript to emphasize this limitation and have included a discussion on the importance of validating the score in settings with highly vaccinated populations.

Regarding the influence of high altitude on the results, we concur that the score appears to work better in high-altitude populations. We have highlighted this finding in our manuscript, noting the implications and the novelty of a scoring system developed for high-altitude populations, as most existing systems have been developed in sea-level populations. We also acknowledge the potential limitations of our score in sea-level populations and have added a section in the discussion to address the need for evaluating the sensitivity, specificity, and other accuracy measures in this subgroup.

We hope that these revisions address your concerns and strengthen our manuscript. We look forward to your feedback and the opportunity to further improve our work.

We have added two paragraphs in the discussion section (page 23 and 24, lines 441 to 482) as follows:

In the presented study, it was observed that the "COVID-19 in-hospital mortality score" demonstrated greater utility in predicting mortality in unvaccinated patients and those residing in higher altitude areas. It is important to note that this finding may be related to specific biological adaptations and environmental factors in these populations [53,54]. However, further research in diverse populations and contexts is required to confirm and better understand these findings.

Higher altitude populations have been shown to exhibit a lower rate of COVID-19 related mortality in some studies (31). This could be due to biological adaptations to hypoxia, which may play a role in the body's response to the virus. Additionally, specific environmental factors of high-altitude areas could influence the spread and severity of the disease (32). On the other hand, in unvaccinated populations, the lack of immunity acquired from vaccination might lead to more severe disease outcomes, which could explain the higher utility of the score in these populations.

Nevertheless, it was found that the score exhibited lower discriminative ability when analyzed in patients from Guayaquil's hospital and in women [S2 Table]. This suggests that the score might be less effective in certain population groups and contexts, emphasizing the importance of validating and adapting the score to different populations and settings.

In conclusion, although the "COVID-19 in-hospital mortality score" appears to be particularly useful in unvaccinated patients and those residing in higher altitude areas, further research is necessary to validate and adapt this score to different populations and contexts. This would enhance its applicability and utility in clinical practice and healthcare resource planning.

References:

(31) Arias-Reyes, C., Zubieta-DeUrioste, N., Poma-Machicao, L., Aliaga-Raduan, F., Carvajal-Rodriguez, F., Dutschmann, M., ... & Soliz, J. (2020). Does the pathogenesis of SARS-CoV-2 virus decrease at high-altitude?. Respiratory Physiology & Neurobiology, 103443.

(32) Simbaña-Rivera K, Jaramillo PRM, Silva JVV, Gómez-Barreno L, Campoverde ABV, Novillo Cevallos JF, et al. High-altitude is associated with better short-term survival in critically ill COVID-19 patients admitted to the ICU. PLoS One. 2022;17(3):e0262423.

CONCLUSION

The objective was to develop a predictive model for mortality in patients hospitalized due to COVID-19. In that sense, the conclusion should answer the objective addressing the usefulness and limitations of the model developed.

ANSWER: 

Thanks for your comment. We recognize the importance of addressing the objective of our study in the conclusion. We will ensure that the conclusion explicitly discusses the utility and limitations of the predictive model for mortality in hospitalized patients with COVID-19.

We have added this text to the abstract (page 4, lines 58 to 61):

A statistically significant Cox regression model with strong discriminatory power and good calibration was developed to predict mortality in hospitalized patients with COVID-19, highlighting its potential clinical utility.

… and conclusion (page 28, lines 586 to 595): 

Based on the statistical significance of the Cox regression model, its discriminatory ability (AUC-ROC), and calibration, the results suggest that the developed model is a valuable tool for predicting mortality in hospitalized COVID-19 patients. The model demonstrates robust statistical performance, with significant associations between the predictor variables and mortality outcomes. Furthermore, the model exhibits good discrimination, as evidenced by a high AUC-ROC value. The calibration analysis confirms the model's ability to accurately estimate mortality probabilities. These findings collectively indicate the potential clinical utility of the model in identifying individuals at higher risk of mortality and informing appropriate interventions. 

 

OTHER COMMENTS FROM REVIEWER 3

Reviewer #3: The cited website link for Ref #1 is broken and number of deaths due to are inaccurate (https://covid19.who.int). Please fix it.

ANSWER: 

Thank you. We have corrected: 

World Health Organization (WHO) has reported a global total of 767,984,989 confirmed cases of COVID-19 caused by Severe acute respiratory syndrome coronavirus 2 (SARS-CoV-2), with 6,943,390 deaths. Additionally, as of 13 June 2023, a total of 13,397,292,784 vaccine doses have been administered worldwide (1).

Reference:

(1) WHO. WHO Coronavirus (COVID-19) Dashboard With Vaccination [Internet]. World Health Organization. 2021. Available from: https://covid19.who.int/

---

## [Editor Report · Decision Letter 2]

21 Jun 2023

Development and validation of a scoring system to predict mortality in patients hospitalized with COVID-19: a retrospective cohort study in two large hospitals in Ecuador

PONE-D-22-16810R2

Dear Dr. Dueñas-Espín,

We’re pleased to inform you that your manuscript has been judged scientifically suitable for publication and will be formally accepted for publication once it meets all outstanding technical requirements.

Kind regards,

Alonso Soto, PhD

Academic Editor

PLOS ONE
---

## [Editor Report · Acceptance letter]

7 Jul 2023

PONE-D-22-16810R2 

Development and validation of a scoring system to predict mortality in patients hospitalized with COVID-19: a retrospective cohort study in two large hospitals in Ecuador 

Dear Dr. Dueñas-Espín:

I'm pleased to inform you that your manuscript has been deemed suitable for publication in PLOS ONE. Congratulations! Your manuscript is now with our production department. 

Kind regards, 

on behalf of

Dr. Alonso Soto 

Academic Editor

PLOS ONE